**Data Availability Statement:** All relevant data are included within the manuscript. The dataset is available upon request in the LSHTM Data

# Challenges to pre-migration interventions to prevent human trafficking: Results from a before-and-after learning assessment of training for prospective female migrants in Odisha, India

**Nicola Suyin Pocock**[1]*, **Ligia Kiss**[1,2], **Mamata Dash**[3], **Joelle Mak**[1], **Cathy Zimmerman**[1]

**1** Gender Violence & Health Centre, Department of Global Health & Development, London School of Hygiene & Tropical Medicine, London, United Kingdom, **2** Institute for Global Health, Faculty of Population Health Science, University College London, Bloomsbury, London, United Kingdom, **3** ASTITWA Gender Resource Centre, Bhubaneswar, India

* nicola.pocock@lshtm.ac.uk, nicolapocock@gmail.com

## Abstract

### Background

Awareness-raising and pre-migration training are popular strategies to prevent human trafficking. Programmatic theories assume that when prospective migrants are equipped with information about risks, they will make more-informed choices, ultimately resulting in safe migration. In 2016, India was estimated to have 8 million people in modern slavery, including those who migrate internally for work. Work in Freedom (WiF) was a community-based trafficking prevention intervention. This study evaluated WiF's pre-migration knowledge-building activities for female migrants in Odisha to prevent future labour-related exploitation.

### Methods

Pre- and post- training questionnaires were administered to women (N = 347) who participated in a two-day pre-migration training session. Descriptive analysis and unadjusted analyses (paired t-tests, McNemar's tests, Wilcoxon signed ranks tests) examined differences in women's knowledge scores before and after training. Adjusted analyses used mixed effects models to explore whether receiving information on workers' rights or working away from home prior to the training was associated with changes in scores. Additionally, we used data from a household survey (N = 4,671) and survey of female migrants (N = 112) from a population sample in the same district to evaluate the intervention's rationale and implementation strategy.

### Results

Female participants were on average 37.3 years-old (SD 11) and most (67.9%) had no formal education. Only 11 participants (3.2%) had previous migration experience. Most participants (90.5%) had previously received information or advice on workers' rights or working

Compass Repository (https://doi.org/10.17037/DATA.00001789). Owing to the sensitive nature of the data and advice of the Research Data Management team, the de-identified dataset is available upon request to qualifying researchers (at the link above). Code, participant information and consent forms and the structured survey instrument are publicly available in the LSHTM Data Compass Repository.

**Funding:** This evaluation was funded by the Department for International Development (DFID), grant number GB-1-203857-102, awarded to LK and CZ at the London School of Hygiene and Tropical Medicine. The funders did not play a role in the study design, data collection and analysis, decision to publish or preparation of the manuscript. Funding URL: https://devtracker.dfid.gov.uk/projects/GB-1-203857/transactions.

**Competing interests:** The authors have declared that no competing interests exist.

away from home. Compared to female migrants in the population, training participants were different in age, caste and religion. Awareness about migration risks, rights and collective bargaining was very low initially and remained low post-training, e.g. of 13 possible migration risks, before the training, participants named an average of 1.2 risks, which increased only slightly to 2.1 risks after the training (T(346) = -11.64, p<0.001). Changes were modest for attitudes about safe and risky migration practices, earnings and savings. Before the training, only 34 women (10.4%) considered migrating, which reduced to 25 women (7.7%) post-training ($X^2$ = 1.88, p = 0.169)—consistent with the low prevalence (7% of households) of female migration locally. Women's attitudes remained relatively fixed about the shame associated with paid domestic work. Survey data indicated focusing on domestic work did not correspond to regional migration trends, where women migrate primarily for construction or agriculture work.

## Conclusion

The apparent low effectiveness of the WiF short-duration migration training may be linked to the assumption that individual changes in knowledge will lead to shifts in social norms. The narrow focus on such individual-level interventions may overestimate an individual's agency. Findings indicate the importance of intervention development research to ensure activities are conducted in the right locations, target the right populations, and have relevant content. Absent intervention development research, this intervention suffered from operating in a site that had very few migrant women and a very small proportion migrating for domestic work—the focus of the training. To promote better development investments, interventions should be informed by local evidence and subjected to rigorous theory-based evaluation to ensure interventions achieve the most robust design to foster safe labour migration for women.

## Background

### Migration awareness raising & effectiveness

Globally there are an estimated 40.3 million people are victims of modern slavery, with approximately 16 million people in forced labour in the private economy [1]. Modern slavery is an umbrella term for various situations where a person is exploited by others for various forms of gain, and includes bonded or forced labour, debt bondage, sexual exploitation, worst forms of child labour, domestic servitude and human trafficking [2].

Preventing modern slavery, a contested concept [3], has become a salient policy concern. Over the past decade, there has been substantial investment in awareness-raising and pre-migration training as a preferred strategy to prevent modern slavery, including human trafficking [4]. This programmatic approach is likely to be related to the limited implementation risks, large potential population reach, low cost and the political acceptability of awareness raising and training interventions [4]. Recently, 37 States endorsed the UN Call to Action to end forced labour, modern slavery and human trafficking, in which they commit to "develop and publish national strategies that raise awareness and improve understanding of the issues amongst the general public and amongst communities considered vulnerable to exploitation" [5]. Awareness-raising activities operate under the programmatic assumption that if

individuals are equipped with knowledge about risks, labour and migration regulations and documentation and their rights, they will be safer during migration and at work [6]. However, recent calls for better evidence and evaluation of interventions to prevent exploitation have suggested the multi-dimensional nature of the risks of human trafficking beyond individual knowledge and behaviour [7].

Globally, India has the highest number of migrants overseas of all countries, where approximately 17.5 million migrants from India have left for destinations including the Gulf states, America, and other destinations [8]. There were 450 million internal migrants according to the latest census [9]. India is estimated to have the largest number of people in modern slavery in the world, with nearly 8 million persons living in modern slavery [10]. In South Asia, modern slavery research and evaluations are clustered in awareness-raising activities at individual and community outcome levels, with a paucity of research and evaluations examining service provider, legal and industry outcomes [2]. To date, few awareness-raising interventions have been rigorously evaluated, as documentation consists of primarily observational studies and process evaluations, which do not assess effectiveness or impact [2].

Despite substantial investments in pre-migration awareness and knowledge-building strategies, to date, there is little to no evidence on their effectiveness to prevent human trafficking [11–13]. Moreover, we know very little about how the content of pre-departure interventions developed for global and regional implementation might influence participants' knowledge, decisions or actions to mitigate risk during migration. Questions remain about whether and how pre-migration information translates into migration plans, actions or outcomes [14]. However, there is a growing understanding that asymmetric power relationships between migrants and their employers and recruitment agencies, which may prevent individuals from applying the information delivered in training programmes [15].

There is a growing body of knowledge on migration decision-making [16, 17]. Theoretical and empirical enquiries are exploring the mechanisms that influence individuals to migrate, examining factors such as individual life course, expectations about migration, gender roles, and community and family norms. Current work is also considering the migration-related impact of natural disasters and conflict, social capital at people's places of origin and destination, and the role of their social networks [16, 18, 19]. Analyses are beginning to assess the differences in outcomes between migration decisions that are prompted by a shock or acute events and those that are well-planned, and how these may affect the migration destination and type of work [17].

Generally, the objective of an awareness-raising and migration training session is to provide information that will support an individual's decision-making, planning process, and behaviours at destination, with the ultimate aim of lowering a migrants' exposure to migration-related abuses. The content of these types of programmes varies widely. In this paper, we discuss the implementation and short-term learning that resulted from a pre-departure orientation for women in Ganjam, Odisha, India, which operated under the larger umbrella programme, the Work in Freedom by the International Labour Organization (ILO). We examine women's before and after training perceptions, knowledge and plans related to migration.

## Out migration from Odisha, India

Odisha is among the poorest states in India, with an official poverty rate of 32.6% compared to 21.9% across India [20]. Women comprise a third of the state's workforce. The Northwest district of Sundergarh is a recruitment area for live-in domestic workers in India's major cities, particular of women from the Adivasi tribe who are preferred due to perception of being submissive [21]. Exploitation by agents and employers and poor working conditions have been documented in this internal labour migration corridor [21, 22]. This study focuses on Ganjam

district, located in south-eastern Odisha, which has not featured in broader debates in the state about labour migration [20]. Known primarily for male out-migration in textiles, diamond factories and ship-breaking industries in Gujarat, very little is known about women's out-migration from Ganjam.

## Work in Freedom intervention

Work in Freedom (WiF) is a large multi-component intervention managed by the ILO and funded by the Department for International Development (DFID). WiF was implemented in India, Nepal and Bangladesh as source countries, and Lebanon and Jordan as destination countries, between 2013 to 2018. One component of the WiF intervention is the community-based activities that aimed to prevent labour trafficking by enhancing women's autonomy, fostering adoption of 'safe migration' practices and assertion of migrant workers' rights [15]. The India component of WiF focussed on prospective in-country migrants only. The implementing partner (ILO) selected Ganjam as the intervention site.

The WiF community-based interventions in India were conducted via direct outreach to women in local households who were offered pre-migration training, which was delivered by peer educators and NGO staff. The community-based intervention aimed to empower women and adolescent girls to avoid what were considered to be 'risky' labour migration practices and labour arrangements, focusing on the paid domestic work sector [23]. Community training sessions were designed to shift attitudes and practices among prospective migrants, household members, and the wider community by delivering key messages about the importance of women's economic and social contributions, recognizing domestic work as a valid form of work, and highlighting risks that arise during the migration process [24].

Direct outreach by partner organizations to recruit participants for the training targeted prospective women workers and their family members (including husbands or parents), who were invited to participate in at least two 60–90 minute conversations in their home or other convenient location. Conversations focussed on safe versus risky migration practices, as well as broader issues around labour, workers' legal and human rights, migration and gender equality. Women did not need to express an interest in or history of migrating for work to be included in these "pre-decision" sessions. This was a deliberate strategy intended to minimise the stigma attached to programme participation, and to maximise the audience receiving programme messages. Direct outreach sessions were unstructured to permit flexible engagement with women and their families [24].

Women who expressed an interest in migrating for work during direct outreach sessions were invited to attend a two-day pre-migration training. Sessions aimed to support women to make informed decisions about whether, and how, to migrate for work, including modules on self-care, financial literacy, use of technology to ensure wellbeing, and safety in transit and at destination. Sessions were organised at the *Gram Panchayat* (village assembly) level and held once approximately 30 women had signed up per session (multiple sessions were held with different groups of approximately 30 women). Overall, a total of 347 participated in the training and completed pre and post training interviews. The training was implemented by the Self Employed Women's Association (SEWA) and *AAINA*, local NGOs with experience in gender, and labour mobilisation and organisation among women [24].

## Methods

### Data

We used data collected by *AAINA*, the Centre for Women's Development Studies and the London School of Hygiene and Tropical Medicine as part of the Study on the Work in Freedom

Transnational (SWIFT) programme evaluation in Odisha, India. Pre and post training questionnaires to measure differences in attitudes, beliefs, plans, intentions and practices were administered by trained interviewers. Preferred selection criteria for interviewers included fluency in local dialects and Odiya language. The survey instrument was designed by the research team with one member (MD) working closely with local partners to appropriately phrase questions. The instrument was translated to Odiya language, and further revised through pilot testing and reviewed after back-translation to English. Vignettes included in the survey instrument were designed with local context in mind, e.g. asking participants what they would do if a recruitment agent came to their village recruiting for work in Mumbai, but that they had heard agents can deceive people. We interviewed all women participating in the WiF training in Ganjam district between January and March 2016. Women were interviewed immediately before and after the training using the structured survey instrument in S1 File. Women were interviewed in the Odiya language or in local dialects where requested.

Additionally, we used data from a representative household survey with female migrants in the same region to describe the context of implementation and WiF's target group. Female migrants were identified through a census of 4,671 households in 20 probabilistically sampled villages in two blocks of Ganjam District, Odisha between November 2015 and February 2017. We used a multistage sample design, with stratification and clustering.

## Ethics

Informed written consent was obtained for all participants, whom were read participant information sheets by trained field interviewers in Odiya language. Where low literacy prevented participants from signing their name, consent was indicated via thumbprint, a common and acceptable practice in the study setting. Participants were assured that data would remain confidential and anonymised, that participation was entirely voluntary, and that they were free to refuse participation and could withdraw from the study at any time. Informed consent was emphasized as an ongoing process, with no adverse consequences for withdrawal from the study at any time.

Ethics approval was obtained from the London School of Hygiene and Tropical Medicine (LSHTM) (reference no. 8840) and the Centre for Women's Development Studies, Indian Council of Social Science Research.

## Variables

**Awareness scores.**   To develop 'awareness scores', women were asked a single question relevant to a construct. For example, awareness of the benefits of migrating for work was asked as follows: "What would you say are the main benefits of moving away from home to take up work somewhere else?" Women answered free form in their own words, i.e. interviewers did not read out item lists. Items that were included in each construct are shown in Table 1. We summed the number of items women mentioned in their response for each construct, before and after the training, and report mean scores in Table 3. In some instances, vignettes were used to prior to eliciting women's awareness on particular constructs (please see Fig 1 –further examples are listed in S1 File, which includes the overall survey instrument). Benefits of using the vignette approach include the potential to tailor questions to specific local contexts, and depersonalization that encourages participants to reflect beyond their own individual circumstances, which is particularly useful when discussing sensitive topics [25, 26].

**Attitudes.**   For questions reporting attitudes, a five-point Likert scale was used. Following a vignette (please see S1 File for examples), women were read a series of statements about the vignette and asked the extent to which they agreed or disagreed. Descriptive results for

**Table 1. Items included in each construct.**

| Construct | Items included |
|---|---|
| **2.3 Main benefits of moving away for work** | Increased earnings |
| | Better quality work |
| | Greater employment security |
| | Greater independence / autonomy at destination |
| | Better standard of living / quality of life at destination |
| | Escape tensions with spouse |
| | Escape tensions with other family members |
| | Escape community tensions |
| | Escape civic or political unrest / violence |
| **2.4 Main dangers/risks of moving away for work** | Earn too little at destination to cover costs of move |
| | Earn too little at destination to fulfil goals (saving, investment, debt-repayment, etc) |
| | Being cheated by agent |
| | Being cheated by employer |
| | Sexual assault |
| | Physical abuse or assault (not sexual) |
| | Forced / Bonded labour situations |
| | Injury |
| | Illness |
| | Isolation / loneliness |
| | Deterioration of relationship with spouse |
| | Deterioration of relationship with children |
| | Social ostracism / stigma on return home |
| **3.1 Information needed before moving away** | General type of work she will be doing |
| | Specific tasks and responsibilities |
| | Name/contact details of middlemen/contractors/placement agents |
| | Name/address/location of employer and workplace |
| | Whether employer provides accommodation |
| | Whether employer provides meals |
| | Amount of wages per day/ week/ month |
| | Frequency of receiving salary |
| | How salary received (e.g. employer or agent) |
| | Costs she will incur (e.g. accommodation/ food / uniform, etc) |
| | Hours of work per day, days per week expected to work |
| | Hours of rest per day |
| | Number of paid off days per week |
| | Cultural differences at destination (e.g. language spoken/ what is the food like) |
| **3.3 Advantages of accepting the advance** | To cover travel costs |
| | To cover setting-up costs at destination |
| | To meet food and daily costs in village prior to leaving |
| | Strengthen relationship with agent |
| | To avoid taking loan on worse terms than advance |
| **3.4 Dangers or risks of accepting advance** | Increased dependency on the agent |
| | Increased dependency on the employer |
| | Increased chance of being cheated by the agent |
| | Increased chance of being cheated by the employer |
| | Less freedom to leave job and return home if dissatisfied |

*(Continued)*

**Table 1.** (Continued)

| Construct | Items included |
|---|---|
| **3.5 Reasons to have a mobile phone while working away from home** | Make and receive calls to keep in touch with family and friends back home |
| | Send and receive photos/ videos/ messages to keep in touch |
| | Take photo of agent and/ or agent's id |
| | Send photo of agent and/ or agent's id to family member or other trusted person |
| | Take photos of important papers to keep a record / use as evidence in a dispute |
| | Send photo of important papers to family member or other trusted person |
| | Call a relative or friend for help if needed |
| | Call an agency or organisation for help if needed (police, NGO or Gov't helpline, etc.) |
| **3.7 How can migrant reduce the chance of an agent cheating or deceiving her** | Find out the agent's full details (name, address, registration number) |
| | Pass the agent's details to a trusted friend or family member |
| | Take a photo of the agent and/ or agent's id |
| | Send a photo of the agent and/ or agent's id to family member/ other trusted person |
| | Refuse to accept an advance payment from the agent |
| | Have employer pay wages directly, not via agent |
| | Send remittances by bank transfer, not via agent |
| **5.1 Rights of domestic workers** | Set pay (set amount of wages / salary) |
| | Hours of work (no more than eight hours of work in a single day) |
| | Set tasks and responsibilities |
| | Regular payment of wages / salary |
| | Rest periods during the working day |
| | Weekly off (paid) |
| | Enough & appropriate food, when employer/middleman provides (i.e. "live-in" workers) |
| | Appropriate accommodation, when employer/middleman provides (i.e. "live-in" workers) |
| | Safety and security at work (and home, when accommodation is provided) |
| | Medical care arranged and paid for by employer if injured or ill at work |
| | Prior notice of dismissal |
| **5.2 Responsibilities of domestic workers** | Beginning work at the agreed time each day |
| | Completing agreed tasks diligently |
| | Maintaining hygiene at work |
| | Respecting the employer's privacy |
| | Notifying employer if unable to work (due to sickness / family emergency, holiday, etc.) |
| | Giving prior notice of resignation |
| **5.2a Where to open a bank account** | Bank |
| | Post Office |
| | Other |
| **6.2 Reasons to join a domestic workers association** | Opportunities to socialise with other women / women workers |
| | Can find out reliable information about worker's rights |
| | Can ask for help from members / organisers to negotiate with employer about pay and working arrangements (hours, tasks, offs, etc.) |
| | Can ask for help from members / organisers if treated badly or abused |
| | Can take part in organised activities to improve paid domestic workers pay and conditions |
| | Can take part in organised activities to improve paid domestic social status |

(*Continued*)

**Table 1.** (Continued)

| Construct | Items included |
|---|---|
| **8.1 Awareness of livelihood and welfare schemes & 8.3 Planning to apply for livelihood & welfare schemes** | Rashtriya Swasthya Bima Yojana (RSBY)/ Biju Krushak Kalyan Yojana (BKKY) (government universal health insurance schemes), or other healthcare |
| | Indira Awas Yojana (government housing scheme for rural poor) (or other housing) |
| | Job/ labour card |
| | Vocational training programmes |
| | National Rural Employment Guarantee Act (NREGA) (wage employment scheme for unskilled manual work) |
| | Pension (old age, widow, disability) |
| | Land patta (land property document issued by government) |
| | Forest patta (forest property document issued by government) |
| | Swachh Bharat (government scheme to install sanitation and solid waste management in communities) |

attitudes were recoded to a three-point scale (Agree, Neutral, Disagree—for presentation simplicity), and unadjusted and adjusted analyses used a five point scale to avoid loss of information (Table 4).

## Analysis

Data cleaning and analysis were conducted in Stata 14. We restricted analysis to women who were surveyed both before and after the training (N = 347). Descriptive analyses and paired t-tests were used to examine whether there were significant changes in awareness scores before and after the training. Unadjusted analyses for changes in binary outcomes were examined using McNemar's test, with Wilcoxon signed ranks tests used to examine changes in Likert scores before and after the training.

In adjusted analyses, we used mixed effects models to analyse the relationship between whether a participant had ever received information on worker's rights or working away from home, as a primary predictor of before and after training scores. We hypothesized that participants who had received prior information would report higher pre and post training scores,

| 3.7 | **An employment agent came to Sula's village to recruit people to work in Mumbai. Sula was interested in going but she had heard that agents sometimes deceive and cheat people…** **What can Sula do to reduce the chance of the agent cheating or deceiving her?** | |
|---|---|---|
| **_DO NOT SHOW OR READ LIST TO RESPONDENT_** (98 = don't know, 88 = prefer not to say) | | √ all reasons respondent mentions |
| i. | Find out the agent's full details (name, address, registration number) | |
| ii. | Pass the agent's details to a trusted friend or family member | |
| iii. | Take a photo of the agent and / or the agent's id | |
| iv. | Send a photo of the agent and / or agent's id to family member or other trusted person | |
| v. | Refuse to accept an advance payment from the agent | |
| vi. | Have employer pay wages directly, not via agent | |
| vii. | Send remittances by bank transfer, not via agent | |
| ix. | Other: | |

**Fig 1. Vignette example used in the pre-post survey.**

compared to participants who had not received prior information. Linear mixed effects models were used for continuous scores, mixed effects logistic regression models for binary outcomes, and mixed effects ordinal logistic regression models for Likert scores respectively [27]. Mixed-effects models account for the fact that multiple responses from the same participant are more similar (within subjects variation) than responses from other participants (between subjects variation) [27]. Models were adjusted for age and education levels. Coefficients and adjusted odds ratios for whether participants received prior information are reported in the final columns of Tables 3, 4. and 5.

## Results

### Intervention context and WiF's target group: Female migrants in Ganjam, Odisha

Participants were selected from a community with predominantly male migration. Of 4,671 households surveyed in Ganjam District, Odisha, 7% reported female migration compared to 44% reporting male migration. Most of these migrant women were from scheduled castes (SC) (74.1%), and one-fifth (19.8%) were from communities that had been involved in conflicts. Christian women were disproportionally represented among female migrants, with Christian households 2.5 times more likely to have a history of female labour migration. The majority of women migrants interviewed (n = 112) made their migration decisions with the input of others (57.4%), whereas 12.2% decided for themselves. Only 4.3% did not have family consent to migrate. The most important reason for migration was chronic or temporary inability to meet basic needs (73.9%).

Almost all migrant women (80.9%) migrated to work in urban destinations. Women were employed in many different sectors, with the most common including construction (25.3%), domestic work (24%) and agriculture (13.9%). Among the migrant women, 4.35% reported forced labour (95%CI 1.5, 10.3)–as assessed by the ILO indicators.

Women who had migrated perceived they developed new skills (63.4%) and gained increased confidence (17.0%). However, most women had mixed feelings about their migration-related gains and losses. For example, approximately one in five (18.9%) believed that migration had both enhanced their skills but reduced their freedom.

### Recruitment and reach of pre-departure training: Participant characteristics

Women and adolescent girls who participated in training were between the ages of 16 and 75, with a mean age of 37.3 years (SD 11) (Table 2). The majority of women (76.0%) were currently married with an average of 2.7 children (SD 2.5) at the time of the study. Half of the women (53.5%) were from castes designated by the Indian government as Other Backward Castes (OBC) while 23.8% were from castes identified as scheduled castes [28]. All participants (100.0%) were Hindu. The majority (67.9%) had no formal education and over half (55.4%) could not read or write any language. One-fifth (18.7%) of participants could read and write Odiya. Women and girls were mainly agricultural labourers (38.4%) or responsible for household chores (27.1%). Just 11 participants (3.2%) had prior experience of migration, while among married women, 42.3% had husbands who had previously migrated. The median household income among participants was 36,000 Indian Rupees/year (Median Average Duration (MAD) 16,000) (USD 474/year, (MAD) USD 211), mainly from work in casual or day-labour in non-agricultural sectors (45.6%). The majority of women (82.4%) reported that their household had no debt. Most households did not own any land (71.0%) or hire in labour

**Table 2. Participant characteristics for women participating in WiF pre-orientation training (n = 347).**

| | N | Percent |
|---|---|---|
| **Age (N = 346)** | | |
| • 16–24 | 37 | 10.7% |
| • 25–34 | 91 | 26.3% |
| • 35–44 | 124 | 35.8% |
| • 45 or older | 94 | 27.2% |
| **Mean age (SD, range)** | 37.3 (SD 11, range 16–75) | |
| **Marital status (N = 342)** | | |
| • Never married | 39 | 11.4% |
| • Currently married | 260 | 76.0% |
| • Divorced/separated | 11 | 3.2% |
| • Abandoned | 32 | 9.4% |
| **Mean number of children (N = 261) (SD, range)\*** | 2.7 (SD 1.8, range 0–22) | |
| **Caste (N = 340)** | | |
| • Scheduled caste | 81 | 23.8% |
| • Scheduled tribe | 65 | 19.1% |
| • Other Backward Caste | 182 | 53.5% |
| • Other | 12 | 3.5% |
| **Religion (N = 341)** | | |
| • Hindu | 341 | 100% |
| **Education level (N = 343)** | | |
| • No formal education | 233 | 67.9% |
| • Primary school | 45 | 13.1% |
| • Secondary school | 64 | 18.7% |
| • University or equivalent | 1 | 0.3% |
| **Literacy level (N = 343)** | | |
| • Cannot read/write any language | 190 | 55.4% |
| • Can read Odiya | 20 | 5.8% |
| • Can write Odiya | 59 | 17.2% |
| • Can read & write Odiya | 64 | 18.7% |
| • Can read Odiya & other language | 2 | 0.6% |
| • Can write Odiya & other language | 2 | 0.6% |
| • Can read & write Odiya & other language | 6 | 1.8% |
| **Mean no. of languages spoken (SD, range)** | 1.0 (SD 0.3, range 1–3) | |
| **Main occupation (N = 328)** | | |
| • Agricultural labourer | 126 | 38.4% |
| • Undertakes household chores | 89 | 27.1% |
| • Construction labourer | 55 | 16.8 |
| • Other | 58 | 17.7% |
| **Prior experience of migration** | 11 | 3.2% |
| **Husband prior experience of migration (N = 257/260 married women)** | 109 | 42.3% |
| **Main source of household income or subsistence (N = 263)** | | |
| • Casual/daily wage labour agricultural | 103 | 39.2% |
| • Casual/daily wage labour non-agricultural | 120 | 45.6% |
| • Own agricultural business | 17 | 6.5% |
| • Own non-agricultural business | 14 | 5.3% |
| • Self-employed with hired workers agricultural | 3 | 1.1% |
| • Self-employed with hired workers non-agricultural | 2 | 0.8% |
| • Regular salaried employment | 4 | 1.5% |
| **Yearly household income (INR) (N = 341)** | | |

(*Continued*)

**Table 2.** (Continued)

| | N | Percent |
|---|---|---|
| • 0–20,000 | 92 | 27.0% |
| • 21,000–40,000 | 132 | 38.7% |
| • 41,000 or more | 117 | 34.3% |
| **Median yearly household income (median average deviation, range) (INR)** | 36,000 (MAD 16,000, range 0–100,000) | |
| **Outstanding household loans (INR) (N = 335)** | | |
| • No debt | 276 | 82.4% |
| • 1–20,000 | 39 | 11.6% |
| • 21,000 or more | 20 | 6.0% |
| **Mean value of outstanding household loans (SD, range) (INR)** | 3,825 (SD 10,631, range 0–100,000) | |
| **No household land owned (N = 343)** | 243 | 71.0% |
| **Household hires in labour (N = 341)** | | |
| • Yes | 62 | 18.2% |
| • No | 271 | 79.5% |
| • Don't know | 5 | 1.5% |
| • Prefer not to say | 3 | 0.9% |
| **Type of household ration card (N = 344)** | | |
| • None | 41 | 11.9% |
| • Below Poverty Line (BPL) | 130 | 37.8% |
| • Above Poverty Line (APL) | 26 | 7.6% |
| • Antyodaya (ration card) | 20 | 5.8% |
| • Other | 91 | 26.5% |
| • Don't know | 2 | 9.9% |
| • Prefer not to say | 34 | 0.6% |
| **Mean number of household members (SD, range)** | 4.5 (SD 1.9, range 0–15) | |
| **Mean number of household members aged 15 or under (SD, range)** | 1.2 (SD 1.2, range 0–5) | |
| **Ever received information or advice on worker's rights (N = 339)** | 33 | 9.7% |
| **Source of advice (N = 29, 4 missing):** | | |
| • Spouse | 5 | 17.2% |
| • Natal family | 14 | 48.3% |
| • Neighbour or friend | 8 | 27.6% |
| • Other | 2 | 6.9% |
| **Ever received information or advice on working away from home (N = 337)** | 303 | 89.9% |
| **Source of advice (N = 298, 5 missing)** | | |
| • Spouse | 105 | 35.2% |
| • Natal family | 51 | 17.1% |
| • Neighbour or friend | 131 | 43.9% |
| • Other | 11 | 3.7% |
| **Ever received any information or advice on rights or working away from home (N = 337)** | 305 | 90.5% |
| **Enrolment in government or NGO livelihood and welfare schemes** | | |
| • Rashtriya Swasthya Bima Yojana (RSBY)/ Biju Krushak Kalyan Yojana (BKKY) (government universal health insurance schemes), or other healthcare | 205/339 | 60.5% |
| • Indira Awas Yojana (government housing scheme for rural poor) (or other housing) | 108/339 | 31.9% |
| • Job/ labour card | 172/340 | 50.6% |
| • Vocational training programmes | - | - |
| • National Rural Employment Guarantee Act (NREGA) (wage employment scheme for unskilled manual work) | 4/337 | 1.2% |
| • Pension (old age, widow, disability) | 126/336 | 37.5% |
| • Land patta (land property document issued by government) | 192/334 | 57.5% |
| • Forest patta (forest property document issued by government) | 6/313 | 1.9% |
| • Swachh Bharat (government scheme to install sanitation and solid waste management in communities) | - | - |

*10 missing among 271 currently married or divorced/separated women

(79.5%). A tenth (11.9%) of participants reported that their household had no ration card, while 37.8% reported having the Below Poverty Line (BPL) card. Women and girls had an average of 4.5 members in their households (SD 1.9). Just a 9.7% had ever received information or advice on worker's rights. Most women and girls (90.5%) had previously received information or advice about working away from home, mainly from neighbours or friends (43.9%) or spouses (35.2%). Over half (60.5%) of participants were enrolled in a healthcare scheme, such as the Rashtriya Swasthya Bima Yojana (RSBY)/ Biju Krushak Kalyan Yojana (BKKY) and half (50.6%) had a job or labour card.

### Awareness and attitudes towards migration risks and opportunities

While unadjusted analyses indicate that there were statistically significant differences in awareness scores before and after the training, very few items were endorsed for each construct (shown in Table 1) at each time point. Changes in scores at pre- and post-training were not practically significant. For example, participants could name 1.2 (SD 0.9) of the main risks of moving away for work before the training, and after the training they could name an average of 2.1 (SD 1.1) (T(346) = -11.64, p<0.001, unadjusted) of 13 possible dangers (Table 3). Similarly, before the training, participants could cite 2.1 (0.9) of 14 types of information that a migrant would need before moving away, and, after the session, they could describe an average of 3.0 (SD 1.2) (T(346) = -12.32, p<0.001, unadjusted) (Table 3). Before the training, participants said that females should be aged 20.4 years old (SD 4.0) before moving away for work, a response they changed to 19.5 years old (SD 2.6) after the training (T(315) = 3.87, p<0.001, unadjusted). When participants were asked to comment on whether the character in the vignette should accept an advance on her wages, before the training, 58.2% of participants thought she should, compared to after the training when 69.3% thought she should (McNemar $X^2$ = 10.98, p<0.001, unadjusted) (Table 3).

In adjusted analyses, having information on migration before the session was associated with increased awareness of the main benefits of moving away for work (β = 0.37, p<0.001) and awareness of the main dangers or risks (β = 0.49, p = 0.001). There was a marginal relationship between having prior information and accepting an advance on wages (AOR 1.5, CI: 0.71–3.14).

In contrast to their awareness scores, participants had greater percentage changes in attitudes towards migration risks and opportunities after the training (Table 4). For example, pre-training, 40.2% of participants agreed that "if someone known to me offers to help me move away to find work, I can be sure they will not abuse or exploit me" compared to 33.0% afterward, indicating a 7.2% reduction in agreement (z = -2.96, p = 0.003, unadjusted) (Table 4). In adjusted analyses, there was a marginal positive relationship between having prior information and being sure that they would not be abused or exploited if someone offered to help them (AOR 1.42, CI: 0.79–2.54). Before training, 87.9% agreed that "Before departing, it is a good idea to check if there is anyone from this village or GP (doctor) at the destination, and to take their contact details", compared to 96.4% after the training, indicating a 8.5% increase in agreement (z = -5.62, p<0.001, unadjusted) (Table 4).

### Awareness and attitudes towards earnings and savings

There were significant changes in attitudes towards savings in four of seven statements. Pre-training, 64.4% of participants disagreed that "Madhuri [character in the vignette] cannot open a savings account without her employer's permission" compared to 78.0% afterward, indicating a 13.6% increase in disagreement (z = -3.16, p = 0.001, unadjusted) (Table 4). There was little change in agreement with the statement "If Madhuri accepts advances on her wages

**Table 3. Awareness of migration risks, opportunities and migration practices, worker's rights, collective bargaining and welfare schemes among women participating in WiF pre-orientation training* (N = 347).**

| | Range + | Pre-training Mean (SD) | Post-training Mean (SD) | Overall Mean change | Unadjusted T(degrees of freedom) score, p-value | Adjusted (any information) Coefficient [95% CI], SE, p-value, adjusted $R^2$ |
|---|---|---|---|---|---|---|
| **2.3 Main benefits of moving away for work score** | 0–9 | 1.3 (0.7) | 1.8 (0.8) | +0.5 | T(346) = -10.41, p<0.001 | β = 0.37[0.18,0.56] SE:0.09, p<0.001, $R^2$ = 0.02 |
| **2.5 Main dangers/risks of moving away for work score** | 0–13 | 1.2 (0.9) | 2.1 (1.1) | +0.9 | T(346) = -11.64, p<0.001 | β = 0.49[0.21,0.77] SE:0.14, p = 0.001, $R^2$ = 0.01 |
| **2.7 Age girl/women should be before moving for work**** | 10–40 | 20.4 (4.0) | 19.5 (2.6) | -0.9 | T(315) = 3.87, p<0.001 | β = -0.81[-1.72,0.83] SE:0.46, p = 0.075, $R^2$ = 0.00 |
| **2.8 Age boy/man should be before moving for work***** | 10–44 | 19.6 (3.1) | 19.1 (2.3) | -0.5 | T(320) = 2.58, p = 0.010 | β = -0.30[-1.16,0.55] SE:0.44, p = 0.488, $R^2$ = 0.00 |
| **3.1 Information needed before moving away score** | 0–14 | 2.1 (0.9) | 3.0 (1.2) | +0.9 | T(346) = -12.32, p<0.001 | β = 0.33[0.04,0.62] SE:0.14, p = 0.024, $R^2$ = 0.03 |
| **3.3 Advantages of accepting the advance score** | 0–5 | 1.2 (0.6) | 1.5 (0.8) | +0.3 | T(346) = -6.39, p<0.001 | β = 0.17[-0.02,0.37] SE:0.10, p = 0.079, $R^2$ = 0.01 |
| **3.4 Dangers or risks of accepting advance score** | 0–5 | 1.0 (0.6) | 1.5 (0.7) | +0.5 | T(346) = -10.63, p<0.001 | β = 0.06[-0.11,0.23] SE:0.08, p = 0.478, $R^2$ = 0.00 |
| **3.5 Reasons to have a mobile phone while working away from home score** | 0–8 | 1.3 (0.5) | 1.9 (0.8) | +0.6 | T(346) = -12.01, p<0.001 | β = 0.03[-0.15,0.22] SE:0.09, p = 0.732, $R^2$ = 0.01 |
| **3.7 How can migrant reduce the chance of an agent cheating or deceiving her score** | 0–7 | 1.2 (0.5) | 1.8 (0.8) | +0.6 | T(346) = -10.58, p<0.001 | β = 0.15[-0.02,0.33] SE:0.09, p = 0.099, $R^2$ = 0.02 |
| **5.1 Rights of domestic workers score** | 0–11 | 2.1 (0.9) | 2.8 (1.3) | +0.7 | T(346) = -8.89, p<0.001 | β = 0.19[-0.08,0.47] SE:0.14, p = 0.180, $R^2$ = 0.03 |
| **5.2 Responsibilities of domestic workers score** | 0–6 | 1.3 (0.6) | 1.9 (0.9) | +0.6 | T(346) = -9.91, p<0.001 | β = 0.01[-0.18,0.21] SE:0.10, p = 0.890, $R^2$ = 0.00 |
| **5.2a Where to open a bank account score** | 0–2 | 1.1 (0.4) | 1.4 (0.5) | +0.3 | T(346) = -8.16, p<0.001 | β = 0.19[0.07,0.31] SE:0.05, p = 0.001, $R^2$ = 0.01 |
| **6.2 Reasons to join a domestic workers association score** | 0–6 | 0.1 (0.5) | 1 (1.1) | +0.9 | T(346) = -13.58, p<0.001 | β = -0.13[-0.37,0.09] SE:0.12, p = 0.245, $R^2$ = 0.00 |
| **8.1 Awareness of livelihood and welfare schemes score** | 0–9 | 4.3 (1.3) | 5.2 (1.7) | +0.9 | T(346) = -8.85, p<0.001 | β = 0.15[-0.26,0.57] SE:0.21, p = 0.473, $R^2$ = 0.00 |
| **8.3 Planning to apply to livelihood and welfare schemes score** | 0–9 | 2.1 (1.5) | 3.1 (1.5) | +1.0 | T(346) = -12.81, p<0.001 | β = 0.16[-0.30,0.63] SE:0.24, p = 0.494, $R^2$ = 0.00 |
| **3.2 Should accept advance on wages (N, %)^** | - | N = 195/ 335 (58.2%) | N = 237/ 342 (69.3% | +11.1% | McNemar $X^2$ = 10.98, p<0.001 | AOR = 1.50 [0.71,3.14], p = 0.277 |
| **6.1 Familiar with domestic worker association (N, %)^^** | - | N = 27/ 75 (36.0%) | N = 157/ 247 (63.6%) | +27.6% | McNemar $X^2$ = 10.67, p = 0.001 | AOR = 0.25 [0.08,0.72], p = 0.010 |

+range for each score is based on complete number of items in that construct (Table 2) that could be endorsed, and is not based on actual data range endorsed by participants, except for Ages females & males should be before moving, which is based on the actual data range reported

*higher scores = greater awareness

**26 missing pre, 5 missing post

***21 missing pre, 5 missing post

^Pre: 3 don't know, 9 missing. Post: 5 missing

^^Pre: 8 missing, 264 don't know. Post: 9 missing, 85 don't know, 6 prefer not to say

from a middleman or employer, she risks being trapped or cheated", with 53.3% agreeing before and 55.2% agreeing afterward (z = -0.30, p = 0.757, unadjusted).

In adjusted analyses, there was a marginal positive relationship between having prior information and; sharing pass book and ATM codes with employers when opening a bank account (AOR 1.50, CI: 0.82–2.74); opening an account before migration to access savings (AOR 1.54, CI: 0.76–3.51), and; it being safer and less costly to send earnings home by bank transfer than

**Table 4. Attitudes towards migration practices, women's work, paid domestic work, earnings and savings among women participating in WiF pre-orientation training (N = 347).**

| | Agree | | | Neutral | | | Disagree | | | Unadjusted | Adjusted (any information) |
|---|---|---|---|---|---|---|---|---|---|---|---|
| | Pre | Post | % change | Pre | Post | % change | Pre | Post | % change | z score p-value | AOR [95% CI] p-value |
| | N (%) | | | N (%) | | | N (%) | | | | |
| **3.6 Attitudes towards safe and risky migration practices** | | | | | | | | | | | |
| If someone known to me offers to help me move away to find work, I can be sure they will *not* abuse or exploit me* | 134 (40.2) | 110 (33.0) | -7.2% | 60 (18.0) | 60 (18.0) | - | 139 (41.7) | 163 (49.0) | +7.3% | z = -2.96 p = 0.003 | AOR = 1.42 [0.79,2.54], p = 0.235 |
| There is nothing a woman can do to avoid being cheated, exploited, or abused if she migrates for work* | 146 (43.7) | 196 (58.7) | +15.0% | 51 (15.3) | 11 (3.3) | -12.0% | 136 (40.8) | 126 (37.8) | -3.0% | z = -2.56 p = 0.010 | AOR = 1.17 [0.66,2.05], p = 0.580 |
| Before departing, it is a good idea to check if there is anyone from this village or GP at the destination, and to take their contact details^ | 291 (87.9) | 319 (96.4) | +8.5% | 20 (6.0) | 8 (2.4) | -3.6% | 20 (6.0) | 4 (1.2) | -4.8% | z = -5.62 p<0.001 | AOR = 1.12 [0.55,2.28], p = 0.750 |
| Migrant workers should inform a local official before they move away, in case they have any problems at destination (e.g. home GP or Labour office, Village head)* | 287 (86.2) | 325 (97.6) | +11.4% | 22 (6.6) | 5 (1.5) | -5.1% | 24 (7.2) | 3 (0.9) | -6.3% | z = -5.50 p<0.001 | AOR = 1.04 [0.60,1.82], p = 0.870 |
| It is against the law to move to another State in India and take up work&& | 75 (23.4) | 64 (19.9) | -3.5% | 57 (17.8) | 25 (7.8) | -10.0% | 189 (58.9) | 232 (72.3) | +13.4% | z = -3.09 p = 0.002 | AOR = 0.91 [0.49,1.72], p = 0.795 |
| **4.1 Attitudes towards women's work** | | | | | | | | | | | |
| "Woman's work" is not as important as "men's work"** | 157 (47.3) | 149 (44.9) | -2.4% | 15 (4.5) | 8 (2.4) | -2.1% | 160 (48.2) | 175 (52.7) | +4.5% | z = -0.83 p = 0.405 | AOR = 1.12 [0.57,2.20], p = 0.725 |
| Men and women should be paid the same for equivalent work* | 191 (57.4) | 258 (77.5) | +20.1% | 32 (9.6) | 2 (0.6) | -9.0% | 110 (33.0) | 73 (21.9) | -11.1% | z = 4.63 p<0.001 | AOR = 0.79 [0.43,1.47], p = 0.475 |
| Women should not take up employment outside the house## | 154 (47.4) | 111 (34.2) | -13.7% | 53 (16.3) | 34 (10.5) | -5.8% | 118 (36.3) | 180 (55.4) | +19.1% | z = -4.55 p<0.001 | AOR = 1.12 [0.67,1.86], p = 0.648 |
| **4.2 Attitudes towards paid domestic work** | | | | | | | | | | | |
| Sita should feel ashamed to do paid domestic work in someone else's home^^ | 144 (43.6) | 154 (46.7) | +3.1% | 10 (3.0) | 3 (0.9) | -2.1% | 176 (53.3) | 173 (52.4) | -0.9% | z = 0.95 p = 0.341 | AOR = 1.07 [0.56,2.03], p = 0.817 |
| Paid domestic work is work like any other^^ | 192 (58.2) | 202 (61.0) | +2.8% | 10 (3.0) | 6 (1.8) | -1.2% | 128 (38.8) | 122 (37.0) | -1.8% | z = 0.75 p = 0.448 | AOR = 0.73 [0.37,1.43], p = 0.363 |
| Paid domestic workers are servants (Chakrani) **not** workers& | 143 (44.4) | 146 (45.3) | +0.9% | 28 (8.7) | 13 (4.0) | -4.7% | 151 (46.9) | 163 (50.6) | +3.7% | z = -0.35 p = 0.722 | AOR = 0.89 [0.49,1.58], p = 0.693 |
| The work paid domestic workers do is essential# | 232 (70.5) | 261 (79.3) | +8.8% | 31 (9.4) | 15 (4.6) | -4.8% | 66 (20.1) | 53 (16.1) | -4.0% | z = 2.70 p = 0.006 | AOR = 0.61 [0.31,1.17], p = 0.139 |
| Paid domestic workers should have respect# | 238 (72.3) | 283 (86.0) | +13.7% | 20 (6.1) | 8 (2.4) | -3.7% | 71 (21.6) | 38 (11.6) | -10.0% | z = 4.40 p<0.001 | AOR = 0.70 [0.36,1.38], p = 0.310 |
| Paid domestic workers have the same rights as all workers^^^ | 151 (47.3) | 191 (59.9) | +12.6% | 55 (17.2) | 33 (10.3) | -6.9% | 113 (35.4) | 95 (29.8) | -5.6% | z = 3.07 p = 0.002 | AOR = 0.80 [0.46,1.39], p = 0.449 |
| **5.3 Attitudes towards earnings and savings** | | | | | | | | | | | |
| If Madhuri opens a savings account she *must* share her pass book and ATM code with her employer^ | 70 (21.2) | 46 (13.9) | -7.3% | 8 (2.4) | - | -2.4% | 253 (76.4) | 285 (86.1) | +9.7% | z = -3.14 p = 0.001 | AOR = 1.50 [0.82,2.74], p = 0.178 |

*(Continued)*

**Table 4.** (Continued)

| | Agree | | | Neutral | | | Disagree | | | Unadjusted | Adjusted (any information) |
|---|---|---|---|---|---|---|---|---|---|---|---|
| | Pre | Post | % change | Pre | Post | % change | Pre | Post | % change | z score p-value | AOR [95% CI] p-value |
| | N (%) | | | N (%) | | | N (%) | | | | |
| Madhuri cannot open a savings account without her employer's permission** | 97 (29.2) | 71 (21.4) | -7.8% | 21 (6.3) | 2 (0.6) | -5.7% | 214 (64.4) | 259 (78.0) | +13.6% | z = -3.16 p = 0.001 | AOR = 1.05 [0.56,1.96], p = 0.872 |
| Madhuri needs a large sum of money to open a savings account with a bank or post office+ | 136 (42.9) | 126 (39.8) | -3.1% | 62 (19.6) | 39 (12.3) | -7.3% | 126 (39.8) | 152 (48.0) | +8.2% | z = -2.04 p = 0.041 | AOR = 1.35 [0.73,2.47], p = 0.332 |
| Madhuri must pay a fee to open a savings account with a bank or post office&& | 148 (46.1) | 184 (57.3) | +11.2% | 55 (17.1) | 40 (12.5) | -4.6% | 118 (36.8) | 97 (30.2) | -6.6% | z = 2.82 p = 0.004 | AOR = 1.20 [0.68,2.12], p = 0.511 |
| Madhuri can open a bank or post office account *before* leaving to work away from home and access her earnings anywhere in India | 230 (72.3) | 275 (86.5) | +14.2% | 40 (12.9) | 30 (9.4) | -3.5% | 48 (15.1) | 13 (4.1) | -11.0% | z = 6.43 p<0.001 | AOR = 1.54 [0.76,3.13], p = 0.224 |
| It is safer and less costly to send earnings home by bank transfer than sending with a person or agent+++ | 231 (73.6) | 265 (84.4) | +10.8% | 33 (10.5) | 23 (7.3) | -3.2% | 50 (15.9) | 26 (8.3) | -7.6% | z = 4.56 p<0.001 | AOR = 1.66 [0.78,3.51], p = 0.183 |
| If Madhuri accepts advances on her wages from a middleman or employer she risks being trapped or cheated++ | 168 (53.3) | 174 (55.2) | +1.9% | 55 (17.5) | 34 (10.8) | -6.7% | 92 (29.2) | 107 (34.0) | +4.8% | z = -0.30 p = 0.757 | AOR = 1.19 [0.63,2.24], p = 0.576 |

*14 missing (n = 333)

**15 missing (n = 332)

^16 missing (n = 331)

^^17 missing (n = 330)

#18 missing (n = 329)

##22 missing (n = 325)

&25 missing (n = 322)

&&26 missing (n = 321)

^^^28 missing (n = 319)

+30 missing (n = 317)

++32 missing (n = 315)

+++33 missing (n = 314)

an agent (AOR 1.66, CI: 0.78–3.51). Whether participants had ever received information prior to the training was not practically or statistically significant in most cases.

## Awareness and attitudes towards domestic work, worker's rights and collective bargaining

There was little awareness of rights of domestic workers before and after the training, with an average of 2.1 (SD 0.9) rights endorsed before training and 2.8 (SD 1.3) rights endorsed afterward (T(346) = -8.89, p<0.001, unadjusted) (Table 3), of a potential 11 rights that could be endorsed (Table 1). Similarly, participants had very low awareness of reasons to join a domestic workers association, with just 0.1 (SD 0.5) reason endorsed before the training and 1 (SD 1.1) reason endorsed after the training (T(346) = -13.58, p<0.001, unadjusted) (Table 3).

For attitudes towards women's work, there was little change in agreement with the statement "Woman's work is not as important as men's work", with 47.3% agreeing before and 44.9% agreeing after the training (z = -0.83, p = 0.405, unadjusted) (Table 4). There was little change in attitudes towards paid domestic work, which were broadly negative. For example,

**Table 5. Intentions to migrate among women participating in WiF pre-orientation training (N = 347).**

| | Pre-training | Post-training | % change | Unadjusted | Adjusted (any information) |
|---|---|---|---|---|---|
| | N (%) | N (%) | | $X^2$, p-value | AOR [95% CI] p-value |
| **7.1 Thoughts about moving from village to work now (n = 326)^** | | | -2.7% | McNemar $X^2$ = 1.88, p = 0.169 | AOR = 0.77 [0.24,2.45], p = 0.670 |
| • Yes | 34 (10.4) | 25 (7.7) | | | |
| • No | 292 (89.6) | 301(92.3) | | | |
| **7.2 Reasons not to move away for work**\* | | | | | |
| 3. No one to care for children | 111 (38.0) | 131 (43.5) | +5.5% | - | - |
| 4. No one to take care of household duties | 99 (33.9) | 122 (40.5) | +6.6% | - | - |
| 8. Spouse will not give permission | 66 (22.6) | 67 (22.2) | -0.4% | - | - |
| 9. Other family member will not give permission | 47 (16.1) | 45 (15.0) | -1.1% | - | - |
| 6. Own ill-health prevents migration | 19 (6.5) | 24 (8.0) | +1.5% | - | - |
| **7.3 Reasons to move away for work**\* | | | | | |
| 9. To seek better paid employment | 16 (47.0) | 8 (32.0) | -15.0% | - | - |
| 1. Chronic/regular inability to meet basic needs in village | 6 (17.6) | 1 (4.0) | -13.6% | - | - |
| 6. Sudden loss of income | 4 (11.8) | 1 (4.0) | -7.8% | - | - |
| 2. Temporary/seasonal inability to meet basic needs in village | 4 (11.8) | - | -11.8% | - | - |
| 11. To seek better quality employment | 2 (5.9) | 8 (32.0) | -26.1% | - | - |
| 15. Condition of advance/loan agreement | 1 (2.9) | 2 (8.0) | +5.1% | - | - |
| 31. Self-advancement | - | 2 (8.0) | +8.0% | - | - |
| 3. Chronic/regular absence of employment in village & surrounding area | 3 (8.8) | 3 (12.0) | +3.2% | - | - |

^21 missing

\*among n = 292 pre and n = 301 post who do not have thoughts of moving away

\*\*among n = 34 pre and n = 25 post who do have thoughts of moving away

nearly half of participants agreed with the statement "Sita should feel ashamed to do paid domestic work in someone else's home" both before (43.6%) and after (46.7%) the training (z = 0.95, p = 0.341, unadjusted). Similarly, 44.4% agreed before and 45.3% after the training that "Paid domestic workers are servants not workers" (z = -0.35, p = 0.722, unadjusted). Changes in attitudes were not practically or statistically significant for these two statements.

In adjusted analyses of awareness and attitudes towards domestic work, worker's rights and collective bargaining, having received information prior to the training was not practically or statistically significant.

## Intentions to apply to welfare schemes and to migrate

Awareness of livelihood and welfare schemes (or programs) was comparatively better than for other constructs, with participants being aware of an average of 4.3 schemes before (SD 1.3) and 5.2 schemes after (SD 1.7) the training (T(346) = -8.85, p<0.001, unadjusted) (Table 3) of a possible 9 schemes (Table 2). Participants planned to apply for an average of 2.1 livelihood and welfare schemes (SD 1.5) before the training, and 3.1 schemes afterward (SD 1.5) (T(346) = -12.81, p<0.001, unadjusted) (Table 3), i.e. they planned to apply for one additional scheme on average.

Intention to migrate was very low and remained low (decreasing slightly) after the training. Just 34 participants (10.4%) had thoughts of moving away from the village to work before the training, compared to 25 participants afterward (7.7%) (McNemar $X^2$ = 1.88, p = 0.169,

**Table 6. Impressions of training delivery and content among women participating in WiF pre-orientation training (N = 347).**

| 9.1 Impressions of orientation delivery and content | Agree N (%) | Neutral N (%) | Disagree N (%) | Don't know N (%) |
|---|---|---|---|---|
| i. I could follow and understand the information I received* | 333 (98.8) | - | 3 (0.9) | 1 (0.3) |
| ii. I learnt information about the risks and dangers of migration that I didn't know before** | 316 (93.2) | 12 (3.5) | 8 (2.4) | 3 (0.9) |
| iii. I learnt information about how to migrate safely that I didn't know before** | 323 (95.3) | 6 (1.8) | 8 (2.4) | 2 (0.6) |
| iv. I learnt information about workers' rights that I didn't know before*** | 285 (84.8) | 8 (2.4) | 22 (6.6) | 21 (6.3) |
| v. I learnt information about women's rights that I didn't know before^ | 289 (85.6) | 6 (1.8) | 21 (6.2) | 21 (6.2) |
| vi. I learnt information about women's anatomy that I didn't know before* | 264 (78.3) | 17 (5.0) | 27 (8.0) | 29 (8.6) |
| vii. I learnt information about good health that I didn't know before^ | 289 (85.3) | 18 (5.3) | 30 (8.9) | 1 (0.3) |
| viii. I learnt information about finance and banking that I didn't know before* | 249 (73.9) | 32 (9.5) | 50 (15.0) | 6 (1.8) |
| ix. I learnt information about government livelihood / welfare schemes that I didn't know before^^ | 210 (62.7) | 74 (22.1) | 39 (11.6) | 12 (3.6) |
| x. I learnt information about how to use a mobile phone that I didn't know before^^ | 256 (76.4) | 27 (8.1) | 50 (14.9) | 2 (0.6) |
| xi. The information I received is relevant to my situation** | 318 (93.8) | 3 (0.9) | 16 (4.7) | 2 (0.6) |
| xii. The information I received is relevant to the situation of my family members** | 319 (94.1) | 7 (2.1) | 12 (3.5) | 1 (0.3) |
| xiii. I enjoyed taking part in the learning activities** | 332 (98.0) | 6 (1.8) | 1 (0.3) | - |
| xiii. If a woman asked me if she should attend the orientation, I would recommend that she do^ | 318 (94.1) | 13 (3.8) | 7 (2.1) | - |

*10 missing

**8 missing

***9 missing

^9 missing

^^12 missing

unadjusted). Having prior information was not associated with intention to migrate in adjusted analyses (AOR 0.77, CI:0.24–2.45) (Table 5).

Among participants who planned to move for work, half (47.0%) cited seeking better paid employment as a reason, compared to a third (32.0%) after the training. After the training, 8 participants (32.0%) also cited seeking better quality employment as a reason to move for work. Among participants who did not have thoughts of moving away for work, before the training, 38.0% cited that there was no one to care for children as the reason, compared to 43.5% afterward (Table 5). One-fifth (22.6% pre and 22.2% post) cited that spouses would not give permission to migrate for work, while one in six (16.1% pre and 15.0% post) reported that another family member would not give permission to migrate for work.

### Impressions of training delivery and content

The majority of women reported positive impressions of the training session, for example, 98.0% enjoyed taking part in the training, while 94.1% felt that the information received was relevant to themselves and family members (Table 6). Almost all women (98.8%) felt that they followed and understood the information received, while 94.1% would recommend the training to other women. Slightly higher proportions of women reported learning new information about risks and dangers of migration (93.2%) and safe migration (95.3%) compared to new information on worker's rights (84.8%), women's rights (85.6%) and good health (85.3%) that they didn't know before (Table 6).

### Discussion

To place this intervention evaluation in context, it is important to acknowledge the relatively low prevalence of households with a history of female migration in Ganjam. The current

scarcity of female migration was subsequently reflected in the small percentage of training participants who intended to migrate. These low migration intentions suggest that the intervention design would have benefited from drawing on the secondary data analysis or conducting formative research to refine the intervention's geographical focus and inform participant recruitment strategy and programme content, before the roll-out of the activities.

For example, the training could have been designed more specifically to recognise the severe economic distress that affects female migrants in Ganjam and the ways they engage in short term circulatory migration and receive casual wages. Moreover, the training content to promote protection could have built on the evidence showing that women often migrate jointly with others. This pattern is in line with the migration literature which posits that migration expectations usually operate in the context of social norms and gender roles, and that migration decisions and behaviours are largely influenced by families and local patterns [18, 29, 30]. In this sense, the apparent low knowledge gain of the WiF short-duration migration training may be linked to the erroneous assumption that individual changes in knowledge will lead to shifts in social norms or can overcome current norms. Community-based interventions that incorporate local actors are probably better-placed to shift social norms than individual-level training by organisations external to targeted communities.

The household data also indicate marked differences between the WiF training participants and the local population of female migrants. For example, while three in four migrants were from scheduled castes, only one-fourth of training participants were from these castes. At the same time, despite the fact that Christian households were more than twice as likely to have female migrants, all women in the training were from Hindu households. These findings suggest that the most likely prospective migrant women were either not recruited into the programme or they were not attracted by the invitations. Future initiatives should consider the target groups' characteristics and dispositions in the design of the intervention and its implementation strategies.

Findings from this study simultaneously raise some broader questions about pre-migration training itself and whether there is sufficient value in investing in information and awareness-raising sessions that prospective migrants may not be able to use in the context of structural drivers of labour exploitation. It is worth asking whether this set of new knowledge on rights and migration processes, while intrinsically valuable to the women, will, in reality, provide protection against exploitation in the face of the enormous power imbalance between migrant women workers and employers. The narrow focus on individual-level interventions may overestimate an individual's agency in face of very real constraints of poverty, debt and family obligations, the complexity and multi-locality of recruitment networks, official corruption and employers impunity [2]. Women may have minimal levels of control over household decisions or migration decision making with agents and when they reach employers, their welfare and wellbeing is often entirely dependent on them. Employers' rights will generally dominate the rights of domestic workers because of social norms and weak implementation of legislation, where it exists, which means that domestic workers will have few avenues to assert new knowledge in a power relationship that widens as women progress along the path of migration [7].

Will women be able to apply their new perspectives to protect themselves against labour exploitation—which was the primary aim of the intervention? If, ultimately, pre-migration training is deemed sufficiently valuable by other studies, then our findings insist that foundational research and adaptive programming evaluation are needed, before substantial investments are made in these types of training events. Pre-departure orientation alone does little to address cultural norms and expectations, structural problems of limited legislation for workers' rights and even weaker enforcement. Coupled with what little rigorous evidence we have

about the effectiveness of pre-departure training, the utility of such programmes should be debated.

It is worrying that women's feelings of shame about undertaking paid domestic work and the value associated with domestic work remained relatively fixed even after hearing training messages to the contrary. Yet, to a certain extent, there were increases in women's attitudes about domestic workers being worthy of respect and that domestic workers should have the same rights as other workers. As noted in the recommendations from the qualitative evaluation of the WiF programme in Bangladesh, in the design of pre-departure training, care must be taken in the promotion of abstract "rights" that women are unlikely to be able to assert [15]. Even where women's awareness of rights has increased, their inability to enforce them can be psychologically painful and damaging [15].

This study was not intended to assess the training techniques, so we are unable to attribute the weak learning levels to a particular aspect of the programme. However, what is clear is that women did not absorb the key messages. For instance, analyses of pre- and post-learning showed that awareness scores (across all domains) were very low before and remained low even after the training. There were greater changes in attitudes towards women's work but still very modest changes in attitudes about migration practices, paid domestic work and earnings and savings. Ultimately, very few women increased their awareness of migration risks and opportunities, worker's rights and collective bargaining. And, these only improved very slightly after the training. Even women's awareness of their rights changed little.

Despite participants' enjoying the sessions and feeling that they understood the information they received, findings show that the key messages were not absorbed during the training. Clearly, evaluations should not be based solely on how participants "felt" about training content, as is so often the case in donor-funded evaluations. Objective measures of whether information received has been absorbed should be included. Moreover, future evaluations should assess if and how women utilise this information. While pre-departure training programmes are often well-received by participants [14] and liked by donors, this simple pre- and post- session learning assessment suggests that challenges remain for the uptake of training messages that must be addressed. Our findings indicate that even after women are provided information about migration-related risks and suggestions about safe practices, women were unlikely to recall or feel able to apply key messages.

For future investments to support prospective migrant women, intervention-focused research should be conducted to identify, for example, what information is prioritised by women, what teaching techniques will be most effective for which populations, and what skill sets are needed by trainers. Future intervention development research should test the effects of different techniques using, for instance, cognitive interviews with participants to assess their acceptance and uptake of the training format and content. And, inquiries should be made about who might be appropriate training staff. For example, findings from the WiF evaluation in Bangladesh highlighted the substantial social and economic gap between the trainers and participants, which created significant learning challenges. For instance, during the sessions, when returnee migrants in Bangladesh wanted to discuss the violence and abuse that they encountered abroad, these reports made trainers feel ill-at-ease and so these discussions were supressed [15].

After the training, fewer women reported intention to migrate—which some might believe is a positive outcome, although this was not the intention of the intervention. Feedback from the local research team suggests a strong ceiling on the migration of women by family members which was internalized as a gender norm "at the cost of starvation". It is also possible that women were dissuaded by descriptions of risks they had not previously considered. Informal feedback from partner organizations suggests that almost all trainees had not ventured beyond

their block or village, with many unable to imagine migrating beyond their home base to a modern city. Unsurprisingly, women's perceptions of the constraints they faced in their own households did not change over the course of the training. Women reported that they would remain burdened with childcare and household duties, and spouses or family members still would not grant permission to migrate. Such types of change will be difficult to achieve in the short term, as they are often embedded in the wider socio-cultural norms. If future programmes aim to rebalance household roles and equity, greater consideration should be given to activities aimed at shifting social norms among male household members [31] or the wider community—however, these activities should not necessarily be conceived of as anti-trafficking or safer migration interventions. Interventions targeting wider socio-cultural norms should arguably be financed by better resourced, overarching development programs, rather than by limited donor funds for anti-trafficking or safer migration interventions, which ought to be more targeted at intervention points along the migration pathway where the structural challenges need to be confronted.

## Limitations

Pre-post surveys cannot indicate how new knowledge will be applied, so findings are limited to what women learned and recalled immediately after the session. The pre-post single group study design is a non-experimental design without a control group, so changes in scores cannot be causally attributed to the intervention.

The very low prevalence of previous migration among women participants in the pre-departure training (N = 11) meant that we could not compare differences in awareness and attitudes between previous migrants and women with no migration experience. In the Nepal WiF prospective migrants survey, returnee women were no better informed than first-time prospective migrants on key aspects of living and working conditions, or on their rights and responsibilities as migrant workers [32]. Returnees were better positioned to share advice on practical and emotional aspects of migration and working abroad, and that may be the case with returnees in Ganjam as well. No upper age limit meant that some women who were very unlikely to ever migrate (i.e. 27.2% of participants were aged 45 or older) were included in the training. A targeted focus on women who had a definitive plan for migration in the short term, or who already had a job offer, may have yielded better evaluation results. However, the likelihood that women would disclose this information given the potential stigma associated with women's labour migration is unclear. Alternative strategies to encourage such discussions are needed. The very low prevalence of previous migration in our findings reflect the general pattern of low female outmigration in Ganjam district indicated by the household survey, where just 9.3% of households had a female migrant [20]. Social stigma associated with female outmigration may have affected reporting in either the household study or the current evaluation findings.

This study was designed to measure the immediate effect of the training among participants and not the long term impact of the intervention given the limitation of a single group pre-post design [33]. However, given that short-term effects of the training were virtually absent, it seems likely that even these small knowledge gains would disappear in the long-term. More importantly, WiF's rationale was: trainings will lead to increased knowledge and empowerment, which will change migration behaviour, which will protect women from human trafficking. However, the evidence from our study did not even support the first link in this assumed causal pathway. This suggests that learning and adaptation are needed before we can recommend the use of empirical design to evaluate the intervention.

Program evaluation data can be useful to improve intervention design and implementation, including course correction in the face of adverse consequences. Practical constraints of program evaluation data include limited data availability, and information on intervention implementation when there is no accompanying process evaluation, as was the case in the present study [34]. A process evaluation may have enabled us to understand whether the training failed because of poor implementation, program theory (individual focus) or a combination of both of these factors.

## Conclusion

Marginal shifts in awareness in this study raise questions about the utility of pre-departure orientations. Future programming requires process evaluations to examine the strengths and weaknesses of implementation and associations between programme delivery and learning outcomes—including longer-term application of knowledge—which will also reflect on the intervention's underlying theory, i.e., pre-migration knowledge is protective against trafficking. While training activities are politically expedient because they are typically easier to implement and calculate outcomes among attendees (e.g., versus interventions to shift structural drivers of gender-inequality, enforcement of worker's rights, occupational health and safety regulations), making women aware of rights that they cannot enforce can be both misleading and, as findings from the WiF evaluation in Bangladesh indicate, possibly psychologically damaging [15]. Before more funds, efforts and hopes are invested in anti-trafficking interventions based on the belief that women will be able to assert their rights, this field will benefit from foundational research to inform interventions that focus on meso, institutional and macro level structures, i.e. employers, workers, frontline civil servants and policy decision makers, which do not place the burden of "keeping safe" on potential migrants themselves.

## Supporting information

**S1 File. Structured survey instrument.**
(PDF)

## Acknowledgments

We are grateful to Samantha Watson for her role in designing and implementing the study, along with partners at *AAINA*, *SEWA* and the Centre for Women's Development Studies.

## Author Contributions

**Conceptualization:** Ligia Kiss, Mamata Dash, Joelle Mak, Cathy Zimmerman.

**Data curation:** Nicola Suyin Pocock.

**Formal analysis:** Nicola Suyin Pocock, Ligia Kiss, Joelle Mak.

**Funding acquisition:** Ligia Kiss, Cathy Zimmerman.

**Investigation:** Nicola Suyin Pocock, Ligia Kiss, Joelle Mak, Cathy Zimmerman.

**Methodology:** Nicola Suyin Pocock, Ligia Kiss, Mamata Dash, Joelle Mak, Cathy Zimmerman.

**Project administration:** Ligia Kiss, Mamata Dash, Joelle Mak.

**Resources:** Ligia Kiss.

**Supervision:** Ligia Kiss, Mamata Dash, Cathy Zimmerman.

**Validation:** Nicola Suyin Pocock, Ligia Kiss, Joelle Mak, Cathy Zimmerman.

**Writing – original draft:** Nicola Suyin Pocock.

**Writing – review & editing:** Nicola Suyin Pocock, Ligia Kiss, Mamata Dash, Joelle Mak, Cathy Zimmerman.

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
