## [Decision Letter · Decision Letter 0]

7 Apr 2020

PONE-D-19-32152

Interventions to prevent human trafficking and labour exploitation: Results from a before-and-after learning assessment of pre-migration training for prospective female migrants in Odisha, India

PLOS ONE

Dear Dr Pocock,

Thank you for submitting your manuscript to PLOS ONE. After careful consideration, we feel that it has merit but does not fully meet PLOS ONE’s publication criteria as it currently stands. Therefore, we invite you to submit a revised version of the manuscript that addresses the points raised during the review process.

Thank you for your patience as this manuscript was reviewed during a very unusual time in our world. Overall, this is an extremely important contribution to the literature. Both reviewers agree that some edits are needed to clarify the site of the research and methods. Both reviewers provided extensive and detailed comments. I would recommend paying particular attention to the following main points if you choose to revise and resubmit this manuscript:

1) I would recommend re-evaluating the title in light of the findings.

2) The background section would benefit from a broader discussion about migration in India followed by additional details about why evaluation of the intervention was done at Ganjam site, where very little was known about women's migration?

3) Please provide additional details for the methods of the pre-decision and pre-migration training and then use terminology consistently. The results appear to focus on those that participated in the pre-decision training, but it is unclear where (or if) the data from the 30ish participants who participated in the pre-migration training is presented? Some specific questions that arose are: 1) How did participants engage and complete surveys if they were illiterate? 2) Please indicate how many participants engaged in the pre-departure training (page 11, line 272). 3) Provide the definition for the RSBY/BKKY abbreviation on page 11, line 290.

4) Please clarify the use of the word scheme.

5) Page 26, lines 547-550 seem to be new results that were not presented previously. This is important information that should be shared in the results section and reflected on in the discussion.

6) Per PLOS One Guidelines, please make sure to use a page number for all citations that include direct quotes and please report exact p-values for all values greater than or equal to 0.001. P-values less than 0.001 may be expressed as p < 0.001 (https://journals.plos.org/plosone/s/submission-guidelines.#loc-statistical-reporting).

We look forward to receiving your revised manuscript.

We would appreciate receiving your revised manuscript by May 08 2020 11:59PM. To enhance the reproducibility of your results, we recommend that if applicable you deposit your laboratory protocols in protocols.io, where a protocol can be assigned its own identifier (DOI) such that it can be cited independently in the future. For instructions see: http://journals.plos.org/plosone/s/submission-guidelines#loc-laboratory-protocols

We look forward to receiving your revised manuscript.

Kind regards,

Michelle L. Munro-Kramer, PhD, CNM, FNP-BC

Academic Editor

PLOS ONE

Additional Editor Comments (if provided):

Thank you for your patience as this manuscript was reviewed during a very unusual time in our world. Overall, this is an extremely important contribution to the literature. Both reviewers agree that some edits are needed to clarify the site of the research and methods. Both reviewers provided extensive and detailed comments. I would recommend paying particular attention to the following main points if you choose to revise and resubmit this manuscript:

1) I would recommend re-evaluating the title in light of the findings.

2) The background section would benefit from a broader discussion about migration in India followed by additional details about why evaluation of the intervention was done at Ganjam site, where very little was known about women's migration?

3) Please provide additional details for the methods of the pre-decision and pre-migration training and then use terminology consistently. The results appear to focus on those that participated in the pre-decision training, but it is unclear where (or if) the data from the 30ish participants who participated in the pre-migration training is presented? Some specific questions that arose are: 1) How did participants engage and complete surveys if they were illiterate? 2) Please indicate how many participants engaged in the pre-departure training (page 11, line 272). 3) Provide the definition for the RSBY/BKKY abbreviation on page 11, line 290.

4) Please clarify the use of the word scheme.

5) Page 26, lines 547-550 seem to be new results that were not presented previously. This is important information that should be shared in the results section and reflected on in the discussion.

6) Per PLOS One Guidelines, please make sure to use a page number for all citations that include direct quotes and please report exact p-values for all values greater than or equal to 0.001. P-values less than 0.001 may be expressed as p < 0.001 (https://journals.plos.org/plosone/s/submission-guidelines.#loc-statistical-reporting).

We look forward to receiving your revised manuscript.

Journal Requirements:

2. Please include additional information regarding all questionnaires used in the study and ensure that you have provided sufficient details that others could replicate the analyses. For instance, if you developed a questionnaire as part of this study and it is not under a copyright more restrictive than CC-BY, please include a copy, in both the original language and English, as Supporting Information.

Reviewers' comments:

Reviewer's Responses to Questions

**Comments to the Author**

1. Is the manuscript technically sound, and do the data support the conclusions?

Reviewer #1: Partly

Reviewer #2: No

2. Has the statistical analysis been performed appropriately and rigorously? 

Reviewer #1: I Don't Know

Reviewer #2: Yes

3. Have the authors made all data underlying the findings in their manuscript fully available?

Reviewer #1: Yes

Reviewer #2: Yes

4. Is the manuscript presented in an intelligible fashion and written in standard English?

Reviewer #1: Yes

Reviewer #2: Yes

5. Review Comments to the Author

Reviewer #1: This manuscript is REALLY important and definitely needs to be published, with some revisions. Hopefully work like this can happen in Western countries like the US and UK where "education" and "awareness" programming is a major (and largely fruitless) effort as well.

Sentence spanning line 122 - 123, citation?

Line 239, insert comma after "binary outcomes"

Line 273 - 4: "The majority of females (76.0%) were currently married with an average of 2.7 children (SD 2.5)." Tense disagreement, also weird to use an adjective as a noun (women or female migrants, but not just females); can change to "the majority of women were, at the time of the investigation, married with..."

Line 275: pls footnote (if journal allows) to link reader to more information about these terms, as "scheduled caste" and "OBC" (especially) sound awful and are unfamiliar to most readers.

Line 281 - 2: "The median household income among participants was 36,000 Indian Rupees/year (MAD 16,000)" contextualise this for readers in some way. comparison to the euro? or what is considered the poverty level in that area? or something else. Also, spell out "MAD" (mean average deviation), and all analytical terms for first use.

Line 290: RSBY/BKKY, define

Table 2: Antyodaya, define; define or contextualise "patta" and all non-English language terms

Line 340: "In adjusted analyses, there was a marginal relationship between..." describe the marginal relationship (positive or negative); same at line 372.

Line 391-2:"Changes in attitudes towards paid domestic work varied." the subsequent statements don't convey variance, but nonvariance in pre- and post-intervention attitudes.

Line 403: scheme =? plan

is this a legal process?

the term "scheme" has negative connotations in Western English (at least, in the USA). but in context, I think this is a normalized term. Please explicitly define its application/use for this manuscript.

Line 412: "Intention to migrate was very low and decreased after the training." seems like it was low and stayed low; it didn't statistically significantly lower.

Lines 470- 472: "...the apparent low effectiveness of the WiF short-duration migration training may be linked to the assumption that individual changes in knowledge will lead to shifts in social norms." and Lines 492 - 493: "The narrow focus on individual-level interventions may overestimate an individual’s agency..." Are pretty massive. IE, they are SUPER important and why this manuscript needs to be published. Please find a way to make these messages bolder. For example, you can title the section so it's even more obvious. "Awareness Minimally Effective" or some such title. These are so quotable, make it easy for readers to find!

Lines 504-5: "Pre-departure orientation alone does little to address CULTURAL NORMS AND EXPECTATIONS, structural problems of lack of workers’ rights, and enforcement of these." insert capitalized text as this was covered in the results.

line 522: "...learning showed that awareness scores were very..." be specific about awareness please; awareness of migrant worker rights and exploitation risks? Or capitalize the titles of the scores, "Awareness" "Attitudes" etc

Lines 539 - 547: while you've commented that the training was probably poorly targeted and not reflective of who is actually migrating for work, you've also noted that awareness trainings are probably insufficient for increased protection again exploitation. could you also call for more research examining what target populations say they need to be protected? examining how cultural norms and expectations impact actions, in conjunction with education/information?

lines 547 - 550: i don't recall seeing this reported in the results. should not share new info in the discussion section. this is important, and should be offered in results.

Line 566-7: "...—however, these activities should not be conceived of as anti-trafficking or safer migration interventions." actually these could be anti-trafficking and safer migration interventions; just more complex to measure impact. such programs would need to be present for years/generations, and sustainable. and their impact should be measured beyond trafficking and migration. (outcome e.g., depression, physical health, alcoholism, IPV, etc).

Alternatively, if you do not see cultural and structural reorganization as anti-trafficking work, please explain why. This manuscript is not necessarily focused on this, but leans into the issue by making a clear case that "awareness raising" activities are insufficient for trafficking prevention. With this statement, you've opened the door for a brief discussion.

lines 571 - 9: is this new information? please present in results.

Line 575: "that" should be "than"

Line 593: change "it is very likely" to "it seems likely"

Line 610: "...psychologically damaging." the data doesn't make the case for this. if you have this data, please share in the results. what you do have data for is that the awareness programming had little to no impact on intention to migrate for work.

Reviewer #2: Overall, I find the topic on pre-training for migrant laborers very compelling - this is an important area of research. Additionally, understanding migration from India is important. However, there is a delink between the title's promise and the article's analysis between pre-migration training and human trafficking. Different countries have different standards as to what forms of labor violations arise to the level of trafficking. Therefore, it would strengthen the overall article if this was clarified a bit more. It appears that much of the questions focus on labor rights and labor violations. But because one experiences a labor violation, does not necessarily make that person trafficked. The paper conveys that overall, the site of study does not have a very high rate migration, the reviewer is left wondering why then, is this an important site of analysis? This would have been strengthened if information on trafficking from the region is provided. Therefore, the significance of analyzing pre- and post-test data from a pre-training for labor migrants in this particular region is very unclear. While this is an important topic, there are major revisions that need to be done to strengthen the paper’s overall contributions to the field of studies in social sciences contributing to migration and human trafficking. Longterm, it would strengthen the study if the a longitudinal study is conducted, where the migrants who received the training were followed overtime. The following are intended to strengthen the paper:

Strengthen overall flow and transitions in the paper:

Page 3 – Paragraph 1: Create a transition from paragraph one into the second paragraph, starts at line 79, “India is estimated…” The sudden shift from training to India is sudden and could use a little set up in the first paragraph.

Provide more contextual information:

Page 3 – Paragraph 2: It would strengthen the overview of India, if the following were accomplished efficiently: 1) what do we know about Indian migration? 2) What do we know about Indians who are trafficked abroad? This would help to tighten the overall direction of the paper which is leading the reader towards examining migration, training and human trafficking. And also if there was a footnote clarifying the author’s understanding of the highly contested terminology, “modern day slavery.” Here it might be useful to draw upon Julia O’Connell Davidson.

Consider reorganizing:

Page 4 – Paragraph on line 97, “There is a growing body of knowledge”. This paragraph should be merged into the following paragraph, “Generally, the objective” on line 107. Otherwise, this paragraph referenced is out of place. It may be beneficial to move it somewhere after the topic sentence. As the authors have introduced at this point that there is a link between awareness raising trainings and “decision-making.”

Provide more clarifications:

Page 5, Section on Work in Freedom intervention – It appears that WIF prioritizes women migrants. It is conveyed that 30 women signed up, yet the participants analyzed was an n of 347, therefore, clarify this overall difference.

Page 4, section on “Out migration from Odisha, India,” – It is unclear why this specific state is important. It would strengthen this section if there was an overall set up of out migration from India broadly, and what the data is regarding migrants from Odisha specifically and how they experience trafficking. What does the research generally show? And if there is very little known, including known cases of trafficked migrants from Odisha, then why should this be a site for the analysis? Are there trainings in Odisha that exist? This would then help to transition into the next section on Work in Freedom Intervention. Otherwise, it is a sudden transition from Odisha India to Work in Freedom Intervention.

Strengthen the methods:

Page 6, Methods, line 184: Cite literature regarding the surveys. Were the researchers involved in the overall design of the pre- and post-test? If yes, then clarify what sources/instrument models were utilized to design the test. When did the post-test occur and what were the decisions regarding this post-test? Also, it is conveyed that participants were interviewed. Provide more information about the recruitment for interviews, and the type of interview that took place (however, I am a bit confused because later in the limits it is stated that the study is limited because it only focuses on pre- and post-test therefore, who was interviewed). It is stated “all women” – was this 30 participants as described in the discussion about WIF or more? Also cite literature for the methodology. Clarify whether all participants literate? What was the vignette participants read or were read? What are the strengths and limits of analyzing data from a program evaluation? Were the researchers a part of the program evaluation or collecting data separate to it? How were decisions made about the design of the pre and post-test. On page 6, it is conveyed that in addition to the test, interviews were conducted. Overview of interview participants and analysis applied to the interviews needs to be included in the study.

Overall, there were indications about participants perceptions beyond the test instruments. It would strengthen the study if these voices were brought more clearly into the study. If interviews did occur, then quotes from participants would strengthen the overall article. If no interviews occurred, then references to interviews need to be clarified. Additionally, if possible, quotes from participants if there were opportunities to fill-out open-ended questions.

Although the language is provided as dialects from various regions, this needs to be very specific. What language were the pre-tests and post-tests done in. Were they translated for the research team?

Table 2. Provide a brief explanation as to why 31 participants are missing. It was also conveyed that interviews were conducted? Was it with 347 participants? Clarify if the demographics data for interviewees and questionnaire participants are the same.

Later on p. 9 it is described that vignettes were read to participants. Explain the logic in the methodology of vignettes.

Strengthen the analysis:

Page 7 , awareness scores – cite literature utilized for decision making regarding awareness scores or cite tools.

Clarify unknowns:

Page 9, Table 1 – explain the acronyms.

Literature review:

The article should have a section on the literature on labor, migration and human trafficking; human trafficking trainings; and the specificity of pre-migration training. This section will help the reader to understand the specific intellectual contribution the authors are making to this dynamic literature. Suggested scholars: Claire Renzetti, Jordan Greenbaum, Karen Albright, Rhacel Salazar Parrenas, Andrijasevic, Lisa Kaida, Elena Shih, Carol Upadhya, Bincy Wilson, Annie Fukushima.

Reorganize:

Participant overview: Page 11 – Participant characteristics- consider moving this section before discussing the analysis. I wanted to know more about the who and recruitment strategies before getting to the analysis.

Analysis: Page 10, - cite literature regarding mixed effects models and who is being drawn upon to inform the own analysis.

Results summary: The results section is organized to describe: the participants; awareness and attitudes towards migration risks and opportunities; awareness and attitudes towards domestic work, worker’s rights and collective bargaining; intentions to apply to welfare schemes and migrate; and impressions of training and delivery and content.

Offer clarification:

Page 14 – acceptance of advance in wages improved after training – it is unclear how this question is about trafficking. It appears to be more about labor violations.

Table 3 – It is unclear as to why there is a focus on domestic workers, when the participant characteristics are that they work a range of industries. This needs to be clarified.

Contextualize information:

Decontextualized information - Page 21 – “In adjusted analyses of awareness and attitudes towards domestic work, worker’s rights and collective bargaining, having received information prior to the training was not practically or statistically significant.” This sentence needs more context and does not really make sense. Explain what is a welfare scheme and the significance. It is unclear why applying for welfare schemes is coupled with intentions to migrate. These are quite different issues that in themselves need analysis and context.

Consider reorganizing to move the discussion of “intentions to migrate on p. 21 earlier to attitudes towards migrations and risks.

Strengthen discussion:

On line 486, page 24, I would have strengthened the discussion if knowledge about rights did not change, then what would have improved this? That is, if the over-emphasis on individuals does not work, then what would have worked to elevate collective understanding about rights? Also, it appears that the trainings deter some prospective migrants from migrating. If this is the case, it appears the program is creating more informed migrants, yet, also deterring interest in migratory processes. It would benefit the article to have more of a discussion about this and provide more context information as to why a case for migration is important (or deterring migration).

Information needs to be contextualized and the methods revisited. On p. 26, it is stated that “when returnee migrants wanted to discuss the violence and abuse that they encountered abroad, these reports made trainers feel ill-at-ease and so these discussions were suppressed.” How do the researchers know what the migrants wanted to discuss? This needs to be clarified – was it in the surveys? Or during the interviews?

Conclusion needs development: the conclusion needs to be strengthened. By analyzing the pre-test and post-test, it seems the overall argument is that the pre-migration training does not work for anti-trafficking efforts. And that there is an overall emphasis on individualism. However, it is unclear how this link is made as the overall analysis and findings do not support this conclusion. It appears to be more of an analysis of the pre- and post-test of pre-departure trainings and attitudes towards migration and work. The link to trafficking is unclear and needs to be strengthened.

6. PLOS authors have the option to publish the peer review history of their article (what does this mean?). If published, this will include your full peer review and any attached files.

Reviewer #1: No

Reviewer #2: No

---

## [Author Response · Author response to Decision Letter 0]

25 Jun 2020

Thank you to the Editor and two Reviewers for your feedback on the manuscript, which helped us to clarify several important points about this study. We are particularly grateful that you took the time to offer detailed comments in the midst of a global pandemic. Please find our responses below.

Editor Comments to the Author

Thank you for your patience as this manuscript was reviewed during a very unusual time in our world. Overall, this is an extremely important contribution to the literature. Both reviewers agree that some edits are needed to clarify the site of the research and methods. Both reviewers provided extensive and detailed comments. I would recommend paying particular attention to the following main points if you choose to revise and resubmit this manuscript:

1) I would recommend re-evaluating the title in light of the findings.

Thank you for the suggestion. We have revised in light of the findings, to ‘Challenges to pre-migration interventions to prevent human trafficking: Results from a before-and-after learning assessment of training for prospective female migrants in Odisha, India’.

2) The background section would benefit from a broader discussion about migration in India followed by additional details about why evaluation of the intervention was done at Ganjam site, where very little was known about women's migration?

We have included more information about migration in India, and a sentence in the WiF intervention section that the implementing partner (ILO) selected Ganjam as the intervention site. The research team had no role in the selection of study sites. It is unclear to us why this site was selected when in hindsight it was perhaps unsuitable. This is one of the key messages of the paper (elaborated in the Discussion and Conclusion) – that formative research should inform these decisions, to ensure that funds are not wasted.

3) Please provide additional details for the methods of the pre-decision and pre-migration training and then use terminology consistently. The results appear to focus on those that participated in the pre-decision training, but it is unclear where (or if) the data from the 30ish participants who participated in the pre-migration training is presented? Some specific questions that arose are: 1) How did participants engage and complete surveys if they were illiterate? 2) Please indicate how many participants engaged in the pre-departure training (page 11, line 272). 3) Provide the definition for the RSBY/BKKY abbreviation on page 11, line 290.

Apologies, we have clarified that it was 30 women per session –

‘Sessions were organised at the Gram Panchayat (village assembly) level and held once approximately 30 women had signed up per session (multiple sessions were held with different groups of approximately 30 women). Overall, a total of 347 participated in the training and completed pre and post training interviews.’

To clarify question 2 (how illiterate women participated in the survey), we included the information about interviews being conducted face-to-face by trained interviewers – 

‘Pre and post training questionnaires to measure differences in attitudes, beliefs, plans, intentions and practices were administered by trained interviewers.’

The definitions for all acronyms are now included in Tables and in the text.

We include a new Ethics sub-section detailing informed consent procedures, and participation in structured survey interviews – 

‘Informed written consent was obtained for all participants, whom were read participant information sheets by trained field interviewers in Odiya language. Where low literacy prevented participants from signing their name, consent was indicated via thumbprint, a common and acceptable practice in the study setting. Participants were assured that data would remain confidential and anonymised, that participation was entirely voluntary, and that they were free to refuse participation and could withdraw from the study at any time. Informed consent was emphasized as an ongoing process, with no adverse consequences for withdrawal from the study at any time.’

4) Please clarify the use of the word scheme.

In Europe, South Asia and Southeast Asia, scheme is a well-known phrase denoting access to e.g. a government welfare scheme. The US equivalent might be ‘benefits program’? We are not keen to change this throughout the manuscript as our primary audience outside the US will know this term, but we include a clarification here for the benefit of US readers - ‘…scheme (or programs)

5) Page 26, lines 547-550 seem to be new results that were not presented previously. This is important information that should be shared in the results section and reflected on in the discussion.

These findings are from the WiF qualitative evaluation in Bangladesh, which is mentioned in the preceding sentence but we have now clarified explicitly. To clarify, this is not new information – 

‘For example, findings from the WiF evaluation in Bangladesh highlighted the substantial social and economic gap between the trainers and participants, which created significant learning challenges. For instance, during the sessions, when returnee migrants in Bangladesh wanted to discuss the violence and abuse that they encountered abroad, these reports made trainers feel ill-at-ease and so these discussions were supressed [15].’ 

6) Per PLOS One Guidelines, please make sure to use a page number for all citations that include direct quotes and please report exact p-values for all values greater than or equal to 0.001. P-values less than 0.001 may be expressed as p < 0.001 (https://journals.plos.org/plosone/s/submission-guidelines.#loc-statistical-reporting).

We have amended as requested.

Review Comments to the Author

Reviewer #1: This manuscript is REALLY important and definitely needs to be published, with some revisions. Hopefully work like this can happen in Western countries like the US and UK where "education" and "awareness" programming is a major (and largely fruitless) effort as well.

Sentence spanning line 122 - 123, citation?

Apologies, we now include a citation.

Line 239, insert comma after "binary outcomes"

Inserted.

Line 273 - 4: "The majority of females (76.0%) were currently married with an average of 2.7 children (SD 2.5)." Tense disagreement, also weird to use an adjective as a noun (women or female migrants, but not just females); can change to "the majority of women were, at the time of the investigation, married with..."

Thank you – rephrased to ‘The majority of women (76.0%) were currently married with an average of 2.7 children (SD 2.5) at the time of the study.’

Line 275: pls footnote (if journal allows) to link reader to more information about these terms, as "scheduled caste" and "OBC" (especially) sound awful and are unfamiliar to most readers.

Unfortunately PLOS does not allow footnotes. We now include a citation here on the origin of the OBC and SC terms. These are officially designated groups in India and these terms are described in the Constitution of India. They are the recognised terms among anyone who works on or in South Asia, and on social justice and human rights. We have added this clarification to the text – 

‘Half of the women (53.5%) were from castes designated by the Indian government as Other Backward Castes (OBC) while 23.8% were from castes identified as scheduled castes’ 

Line 281 - 2: "The median household income among participants was 36,000 Indian Rupees/year (MAD 16,000)" contextualise this for readers in some way. comparison to the euro? or what is considered the poverty level in that area? or something else. Also, spell out "MAD" (mean average deviation), and all analytical terms for first use.

Thank you – we have spelled out MAD and included USD values for comparison.

Line 290: RSBY/BKKY, define

We have defined in the text and in Table 1.

Table 2: Antyodaya, define; define or contextualise "patta" and all non-English language terms

There is no direct translation, Antyodaya is a type of ration card. We believe that this is reflected in the ‘Type of household ration card (N=344)’ header above, but have included (ration card) again here for clarity. We have defined and contextualized all remaining terms in the last cell of Table 2, and included these definitions in Table 1.

Line 340: "In adjusted analyses, there was a marginal relationship between..." describe the marginal relationship (positive or negative); same at line 372.

Thank you, we have clarified these sentences indicating positive marginal relationships.

Line 391-2:"Changes in attitudes towards paid domestic work varied." the subsequent statements don't convey variance, but nonvariance in pre- and post-intervention attitudes.

Thank you for spotting this – we have rephrased the sentence ‘There was little change in attitudes towards paid domestic work, which were broadly negative.’

Line 403: scheme =? plan

is this a legal process?

the term "scheme" has negative connotations in Western English (at least, in the USA). but in context, I think this is a normalized term. Please explicitly define its application/use for this manuscript.

In Europe, South Asia and Southeast Asia, scheme is a well-known phrase denoting access to e.g. a government welfare scheme. The US equivalent might be ‘benefits program’? We are not keen to change this throughout the manuscript as our primary audience outside the US will know this term, but we include a clarification here for the benefit of US readers - ‘…scheme (or programs)’.

Line 412: "Intention to migrate was very low and decreased after the training." seems like it was low and stayed low; it didn't statistically significantly lower.

Thank you – we have rephrased to - ‘Intention to migrate was very low and remained low (decreasing slightly) after the training’.

Lines 470- 472: "...the apparent low effectiveness of the WiF short-duration migration training may be linked to the assumption that individual changes in knowledge will lead to shifts in social norms." and Lines 492 - 493: "The narrow focus on individual-level interventions may overestimate an individual’s agency..." Are pretty massive. IE, they are SUPER important and why this manuscript needs to be published. Please find a way to make these messages bolder. For example, you can title the section so it's even more obvious. "Awareness Minimally Effective" or some such title. These are so quotable, make it easy for readers to find!

Thank you - we have included these two salient points in the Abstract conclusion so that they are emphasised more heavily.

Lines 504-5: "Pre-departure orientation alone does little to address CULTURAL NORMS AND EXPECTATIONS, structural problems of lack of workers’ rights, and enforcement of these." insert capitalized text as this was covered in the results.

Thank you for the suggestion – we have rephrased as suggested.

line 522: "...learning showed that awareness scores were very..." be specific about awareness please; awareness of migrant worker rights and exploitation risks? Or capitalize the titles of the scores, "Awareness" "Attitudes" etc

Thank you – we have clarified that awareness was low across all domains - ‘For instance, analyses of pre- and post-learning showed that awareness scores (across all domains) were very low before and remained low even after the training.’ 

Lines 539 - 547: while you've commented that the training was probably poorly targeted and not reflective of who is actually migrating for work, you've also noted that awareness trainings are probably insufficient for increased protection again exploitation. could you also call for more research examining what target populations say they need to be protected? examining how cultural norms and expectations impact actions, in conjunction with education/information?

Thank you for the suggestion. We believe that these points are covered more broadly in the Discussion, e.g. first paragraph outlining the need for formative research to target the intervention activities appropriately based on demand. We prefer to keep the remaining paragraphs as tightly related to what would have improved the WiF intervention and evaluation as possible.

lines 547 - 550: i don't recall seeing this reported in the results. should not share new info in the discussion section. this is important, and should be offered in results.

These findings are from the WiF qualitative evaluation in Bangladesh, which is mentioned in the preceding sentence but we have now clarified explicitly. To clarify, this is not new information – 

‘For example, findings from the WiF evaluation in Bangladesh highlighted the substantial social and economic gap between the trainers and participants, which created significant learning challenges. For instance, during the sessions, when returnee migrants in Bangladesh wanted to discuss the violence and abuse that they encountered abroad, these reports made trainers feel ill-at-ease and so these discussions were supressed [15].’ 

Line 566-7: "...—however, these activities should not be conceived of as anti-trafficking or safer migration interventions." actually these could be anti-trafficking and safer migration interventions; just more complex to measure impact. such programs would need to be present for years/generations, and sustainable. and their impact should be measured beyond trafficking and migration. (outcome e.g., depression, physical health, alcoholism, IPV, etc).

Alternatively, if you do not see cultural and structural reorganization as anti-trafficking work, please explain why. This manuscript is not necessarily focused on this, but leans into the issue by making a clear case that "awareness raising" activities are insufficient for trafficking prevention. With this statement, you've opened the door for a brief discussion.

Thank you - we have rephrased to elaborate on this position further and caveated the initial sentence – 

‘however, these activities should not necessarily be conceived of as anti-trafficking or safer migration interventions. Interventions targeting wider socio-cultural norms should arguably be financed by better resourced, overarching development programs, rather than by limited donor funds for anti-trafficking or safer migration interventions, which ought to be more targeted at intervention points along the migration pathway where the structural challenges need to be confronted..’

lines 571 - 9: is this new information? please present in results.

Line 575: "that" should be "than"

Thank you – we have changed to ‘than’. This is not new information – there is a citation in the text indicating that these findings apply to Nepal, with potential relevance for the current findings in Ganjam, India –

‘In the Nepal WiF prospective migrants survey, returnee women were no better informed than first-time prospective migrants on key aspects of living and working conditions, or on their rights and responsibilities as migrant workers [32]. Returnees were better positioned to share advice on practical and emotional aspects of migration and working abroad, and that may be the case with returnees in Ganjam as well.’

Line 593: change "it is very likely" to "it seems likely"

Thank you – we have amended as suggested.

Line 610: "...psychologically damaging." the data doesn't make the case for this. if you have this data, please share in the results. what you do have data for is that the awareness programming had little to no impact on intention to migrate for work.

Thank you – we have included a citation to the WiF evaluation in Bangladesh and caveated this sentence- ‘making women aware of rights that they cannot enforce can be both misleading and as findings from the WiF evaluation in Bangladesh indicate, possibly psychologically damaging [15].’

Reviewer #2: Overall, I find the topic on pre-training for migrant laborers very compelling - this is an important area of research. Additionally, understanding migration from India is important. However, there is a delink between the title's promise and the article's analysis between pre-migration training and human trafficking. Different countries have different standards as to what forms of labor violations arise to the level of trafficking. Therefore, it would strengthen the overall article if this was clarified a bit more. It appears that much of the questions focus on labor rights and labor violations. But because one experiences a labor violation, does not necessarily make that person trafficked. The paper conveys that overall, the site of study does not have a very high rate migration, the reviewer is left wondering why then, is this an important site of analysis? This would have been strengthened if information on trafficking from the region is provided. Therefore, the significance of analyzing pre- and post-test data from a pre-training for labor migrants in this particular region is very unclear. While this is an important topic, there are major revisions that need to be done to strengthen the paper’s overall contributions to the field of studies in social sciences contributing to migration and human trafficking. Longterm, it would strengthen the study if the a longitudinal study is conducted, where the migrants who received the training were followed overtime. The following are intended to strengthen the paper:

Strengthen overall flow and transitions in the paper:

Page 3 – Paragraph 1: Create a transition from paragraph one into the second paragraph, starts at line 79, “India is estimated…” The sudden shift from training to India is sudden and could use a little set up in the first paragraph.

Thank you for the suggestion – we have included an opening paragraph citing global modern slavery estimates, which we believe helps the transition to the subsequent paragraph about India. 

Provide more contextual information:

Page 3 – Paragraph 2: It would strengthen the overview of India, if the following were accomplished efficiently: 1) what do we know about Indian migration? 2) What do we know about Indians who are trafficked abroad? This would help to tighten the overall direction of the paper which is leading the reader towards examining migration, training and human trafficking. And also if there was a footnote clarifying the author’s understanding of the highly contested terminology, “modern day slavery.” Here it might be useful to draw upon Julia O’Connell Davidson.

Thank you – we include information on outmigration and internal migration in India, there are no reliable estimates for numbers trafficked abroad. We include a reference to O’Connell Davidson in an earlier sentence clarifying that the concept of modern slavery is contested.

‘Globally, India has the highest number of migrants overseas of all countries, where approximately 17.5 million migrants from India have left for destinations including the Gulf states, Europe and other destinations [8]. There were 450 million internal migrants according to the latest census [9]. 

Consider reorganizing:

Page 4 – Paragraph on line 97, “There is a growing body of knowledge”. This paragraph should be merged into the following paragraph, “Generally, the objective” on line 107. Otherwise, this paragraph referenced is out of place. It may be beneficial to move it somewhere after the topic sentence. As the authors have introduced at this point that there is a link between awareness raising trainings and “decision-making.”

Thank you – we prefer to keep these separate. The paragraph on line 97 is about decision-making independent of awareness-raising training, while the link is discussed in the subsequent paragraph.

Provide more clarifications:

Page 5, Section on Work in Freedom intervention – It appears that WIF prioritizes women migrants. It is conveyed that 30 women signed up, yet the participants analyzed was an n of 347, therefore, clarify this overall difference.

Thank you – we have clarified this sentence – 

‘Sessions were organised at the Gram Panchayat (village assembly) level and held once approximately 30 women had signed up per session (multiple sessions were held with different groups of approximately 30 women). Overall, a total of 347 participated in the training and completed pre and post training interviews.’

Page 4, section on “Out migration from Odisha, India,” – It is unclear why this specific state is important. It would strengthen this section if there was an overall set up of out migration from India broadly, and what the data is regarding migrants from Odisha specifically and how they experience trafficking. What does the research generally show? And if there is very little known, including known cases of trafficked migrants from Odisha, then why should this be a site for the analysis? Are there trainings in Odisha that exist? This would then help to transition into the next section on Work in Freedom Intervention. Otherwise, it is a sudden transition from Odisha India to Work in Freedom Intervention.

We believe that we address the importance of this state in the paragraph, citing the poverty rate, the lack of focus on outmigration from Ganjam district in Odisha to other parts of the state, and lack of focus on internal migration for domestic work among women in Odisha. We include citations to research about exploitation by agents and employers in the state. The ILO selected the WiF intervention site without consultation with the evaluation team, as explained in the concluding sentence of the previous paragraph, which helps with the subsequent transition to the WiF intervention description. 

Strengthen the methods:

Thank you for the detailed comments. We break down the comments individually, as there are a large number of questions in each.

Page 6, Methods, line 184: Cite literature regarding the surveys. Were the researchers involved in the overall design of the pre- and post-test? If yes, then clarify what sources/instrument models were utilized to design the test. 

Thank you – we have rephrased –

‘The survey instrument was designed by the research team with one member (MD) working closely with local partners to appropriately phrase questions. The instrument was translated to Odiya language.’ 

We also now include the full survey instrument as Supplementary information. 

When did the post-test occur and what were the decisions regarding this post-test? Also, it is conveyed that participants were interviewed. Provide more information about the recruitment for interviews, and the type of interview that took place (however, I am a bit confused because later in the limits it is stated that the study is limited because it only focuses on pre- and post-test therefore, who was interviewed). 

We include a clarifying sentence – ‘Women were interviewed immediately before and after the training.’

Earlier information on participant recruitment is provided in prior paragraphs on the Work in Freedom intervention –

‘Direct outreach by partner organizations to recruit participants for the training targeted prospective women workers and their family members (including husbands or parents), who were invited to participate in at least two 60-90 minute conversations in their home or other convenient location….

Women who expressed an interest in migrating for work during direct outreach sessions were invited to attend a two-day pre-migration training.’

It is stated “all women” – was this 30 participants as described in the discussion about WIF or more? Also cite literature for the methodology. Clarify whether all participants literate? What was the vignette participants read or were read? 

We now clarify that it is 30 participants/session with multiple sessions in the earlier WiF intervention section – 

‘Sessions were organised at the Gram Panchayat (village assembly) level and held once approximately 30 women had signed up per session (multiple sessions were held with different groups of approximately 30 women). Overall, a total of 347 participated in the training and completed pre and post training interviews.’ 

Literacy levels are part of the Results and are included in Table 2. Further information on vignettes is provided later in the Methods section, where we detail the process of women being read the vignette by interviewers. We describe the methodology and intervention, with appropriate references in that section. The survey instrument was conceived by the research team, and is provided in S1.

What are the strengths and limits of analyzing data from a program evaluation? Were the researchers a part of the program evaluation or collecting data separate to it? How were decisions made about the design of the pre and post-test. On page 6, it is conveyed that in addition to the test, interviews were conducted. Overview of interview participants and analysis applied to the interviews needs to be included in the study.

By interviews, these are survey interviews – we have rephrased to clarify – 

‘Women were interviewed immediately before and after the training using the structured survey instrument in S1.’

As mentioned in the background section, these data are part of a program evaluation (there was no separate data collection). We now include a section in Discussion > Limitations on the benefits and limitations of program evaluation data – 

‘Program evaluation data can be useful to improve intervention design and implementation, including course correction in face of adverse consequences. Practical constraints of program evaluation data include limited data availability, and information on intervention implementation when there is no accompanying process evaluation, as was the case in the present study [34]. A process evaluation may have enabled us to understand whether the training failed because of poor implementation, program theory (individual focus) or a combination of both of these factors.’

Overall, there were indications about participants perceptions beyond the test instruments. It would strengthen the study if these voices were brought more clearly into the study. If interviews did occur, then quotes from participants would strengthen the overall article. If no interviews occurred, then references to interviews need to be clarified. Additionally, if possible, quotes from participants if there were opportunities to fill-out open-ended questions.

Thank you – there were no qualitative interviews. We have clarified in this sentence, as above – 

‘Women were interviewed immediately before and after the training using the structured survey instrument in S1.’

Although the language is provided as dialects from various regions, this needs to be very specific. What language were the pre-tests and post-tests done in. Were they translated for the research team?

As detailed in the Data section, the survey instrument was translated to Odiya language. Interviews were conducted in Odiya or in the participant’s dialect (if spoken by the interviewer). We have clarified –

‘Women were interviewed in the Odiya language or in local dialects where requested.’

Table 2. Provide a brief explanation as to why 31 participants are missing. It was also conveyed that interviews were conducted? Was it with 347 participants? Clarify if the demographics data for interviewees and questionnaire participants are the same.

Thank you – we have checked and unmarried women were mistakenly asked this question in the survey, which accounts for most of the missing responses. We have re-run this analysis for currently/formerly married women and only 10 responses are missing – this is now corrected in the manuscript. As clarified above, survey interviews were conducted, not qualitative interviews.

Later on p. 9 it is described that vignettes were read to participants. Explain the logic in the methodology of vignettes.

Thank you – we now include a sentence on the approach and methodology –

‘Benefits of using the vignette approach include the potential to tailor questions to specific local contexts, and depersonalization that encourages participants to reflect beyond their own individual circumstances, which is particularly useful when discussing sensitive topics [25,26].’

Strengthen the analysis:

Page 7 , awareness scores – cite literature utilized for decision making regarding awareness scores or cite tools.

The structured survey instrument is available in S1 with vignettes in S2 – these are cited in the text.

Clarify unknowns:

Page 9, Table 1 – explain the acronyms.

These are now clarified.

Literature review:

The article should have a section on the literature on labor, migration and human trafficking; human trafficking trainings; and the specificity of pre-migration training. This section will help the reader to understand the specific intellectual contribution the authors are making to this dynamic literature. Suggested scholars: Claire Renzetti, Jordan Greenbaum, Karen Albright, Rhacel Salazar Parrenas, Andrijasevic, Lisa Kaida, Elena Shih, Carol Upadhya, Bincy Wilson, Annie Fukushima.

Thank you for the suggestion. We cite literature on labour, migration and human trafficking awareness raising interventions in the background section and are simultaneously cognisant of the word limitations. We prefer to keep the discussion centred on the evaluation findings, and cite supporting literature here as well.

Reorganize:

Participant overview: Page 11 – Participant characteristics- consider moving this section before discussing the analysis. I wanted to know more about the who and recruitment strategies before getting to the analysis.

Participant characteristics are placed before the analysis in results, which follows in subsequent sections. Recruitment strategies are detailed earlier in the WiF Intervention section.

Analysis: Page 10, - cite literature regarding mixed effects models and who is being drawn upon to inform the own analysis.

Thank you – we now include a citation.

Results summary: The results section is organized to describe: the participants; awareness and attitudes towards migration risks and opportunities; awareness and attitudes towards domestic work, worker’s rights and collective bargaining; intentions to apply to welfare schemes and migrate; and impressions of training and delivery and content.

Offer clarification:

Page 14 – acceptance of advance in wages improved after training – it is unclear how this question is about trafficking. It appears to be more about labor violations.

Acceptance of wage advances is considered risky as it can trap workers in debt bondage situations. As such, it was included in the training. We understand that trafficking is a contested concept in academic circles, and defined differently in legal and policy spaces, but in practice, organisations have deemed there are indicators of risk that can span various levels of labour violations.

Table 3 – It is unclear as to why there is a focus on domestic workers, when the participant characteristics are that they work a range of industries. This needs to be clarified.

Thank you. This is the point of the paper, which we elaborate in the Discussion – the WiF intervention was a community-based intervention focussing on the paid domestic work sector (Background > ‘WiF intervention’). But, participants were women who were very unlikely to migrate alone, with few expressing an interest in doing so before the training. And, according to regional migration trends, women would migrate primarily for construction or agriculture (Abstract > Results, and in Discussion). The decision to focus on domestic work was taken by the WiF implementing partner, the ILO. This decision may not have been based on adequate foundational evidence, and hence, as we elaborate in Discussion, meant that the intervention was probably poorly targeted. 

Contextualize information:

Decontextualized information - Page 21 – “In adjusted analyses of awareness and attitudes towards domestic work, worker’s rights and collective bargaining, having received information prior to the training was not practically or statistically significant.” This sentence needs more context and does not really make sense. 

We prefer to report results succinctly based on analyses alone, with results contextualised in the Discussion. Having prior information about ever working away from home or workers’ rights, did not affect participants attitudes or awareness towards domestic work, workers’ rights, or collective bargaining. 

Explain what is a welfare scheme and the significance. It is unclear why applying for welfare schemes is coupled with intentions to migrate. These are quite different issues that in themselves need analysis and context.

We now include ‘welfare schemes (or programs)’ for the benefit of US readers, who may be unfamiliar with the term ‘scheme’. Awareness (and enrolment) in welfare schemes is important as much migration is driven by financial crises. Among the assumptions of the pre-migration interventions was the belief that welfare schemes can help migrants to ‘remain in place’, which is why it is mentioned in this section on migration intention. 

Consider reorganizing to move the discussion of “intentions to migrate on p. 21 earlier to attitudes towards migrations and risks.

We believe it makes sense for these findings remain here after the overall findings on awareness and attitudes.

Strengthen discussion:

On line 486, page 24, I would have strengthened the discussion if knowledge about rights did not change, then what would have improved this? That is, if the over-emphasis on individuals does not work, then what would have worked to elevate collective understanding about rights? Also, it appears that the trainings deter some prospective migrants from migrating. If this is the case, it appears the program is creating more informed migrants, yet, also deterring interest in migratory processes. It would benefit the article to have more of a discussion about this and provide more context information as to why a case for migration is important (or deterring migration).

We are not sure what would have improved knowledge about rights. As noted in the Conclusion, results may reflect a failure in the intervention assumptions. We discuss how the training possibly deterred women from migrating – 

‘After the training, fewer women reported intention to migrate—which some might believe is a positive outcome, although this was not the intention of the intervention. Feedback from the local research team suggests a strong ceiling on the migration of women by family members which was internalized as a gender norm “at the cost of starvation”. It is also possible that women were dissuaded by descriptions of risks they had not previously considered. Informal feedback from partner organizations suggests that almost all trainees had not ventured beyond their block or village, with many unable to imagine migrating beyond their home base to a modern city.’

Information needs to be contextualized and the methods revisited. On p. 26, it is stated that “when returnee migrants wanted to discuss the violence and abuse that they encountered abroad, these reports made trainers feel ill-at-ease and so these discussions were suppressed.” How do the researchers know what the migrants wanted to discuss? This needs to be clarified – was it in the surveys? Or during the interviews?

These findings are from the WiF evaluation in Bangladesh, which we have now clarified in the text. 

Conclusion needs development: the conclusion needs to be strengthened. By analyzing the pre-test and post-test, it seems the overall argument is that the pre-migration training does not work for anti-trafficking efforts. And that there is an overall emphasis on individualism. However, it is unclear how this link is made as the overall analysis and findings do not support this conclusion. It appears to be more of an analysis of the pre- and post-test of pre-departure trainings and attitudes towards migration and work. The link to trafficking is unclear and needs to be strengthened.

We appreciate the observation of the emphasis on ‘individualism’ and the links between our findings and human trafficking. As stated in the Discussion text below, we note that we are going beyond the intervention learning results and questioning the value of the theory underlying the intervention itself in light of the many intransigent global and local structures that foster exploitation including human trafficking - 

‘Findings from this study simultaneously raise some broader questions about pre-migration training itself and whether there is sufficient value in investing in information and awareness-raising sessions that prospective migrants may not be able to use in the context of structural drivers of labour exploitation. It is worth asking whether this set of new knowledge on rights and migration processes, while intrinsically valuable to the women, will, in reality, provide protection against exploitation in the face of the enormous power imbalance between migrant women workers and employers. The narrow focus on individual-level interventions may overestimate an individual’s agency in face of very real constraints of poverty, debt and family obligations, the complexity and multi-locality of recruitment networks, official corruption and employers impunity [6].’

---

## [Decision Letter · Decision Letter 1]

25 Aug 2020

Challenges to pre-migration interventions to prevent human trafficking: Results from a before-and-after learning assessment of training for prospective female migrants in Odisha, India

PONE-D-19-32152R1

Dear Dr. Pocock,

We’re pleased to inform you that your manuscript has been judged scientifically suitable for publication and will be formally accepted for publication once it meets all outstanding technical requirements.

Kind regards,

Michelle L. Munro-Kramer, PhD, CNM, FNP-BC

Academic Editor

PLOS ONE

Additional Editor Comments (optional): Thank you for taking the time to address all reviewer comments. I look forward to seeing this manuscript in publication.

Reviewers' comments:

Reviewer's Responses to Questions

**Comments to the Author**

1. If the authors have adequately addressed your comments raised in a previous round of review and you feel that this manuscript is now acceptable for publication, you may indicate that here to bypass the “Comments to the Author” section, enter your conflict of interest statement in the “Confidential to Editor” section, and submit your "Accept" recommendation.

Reviewer #1: All comments have been addressed

Reviewer #2: All comments have been addressed

2. Is the manuscript technically sound, and do the data support the conclusions?

Reviewer #1: (No Response)

Reviewer #2: Yes

3. Has the statistical analysis been performed appropriately and rigorously? 

Reviewer #1: (No Response)

Reviewer #2: Yes

4. Have the authors made all data underlying the findings in their manuscript fully available?

Reviewer #1: (No Response)

Reviewer #2: Yes

5. Is the manuscript presented in an intelligible fashion and written in standard English?

Reviewer #1: (No Response)

Reviewer #2: Yes

6. Review Comments to the Author

Reviewer #1: (No Response)

Reviewer #2: Overall, I believe the paper is important and should be published. I found it compelling and there is a ton of interest in pre-departure trainings. The comments were addressed, and if there was disagreement, I found the authors logic made sense.

7. PLOS authors have the option to publish the peer review history of their article (what does this mean?). If published, this will include your full peer review and any attached files.

Reviewer #1: No

Reviewer #2: No

---

## [Editor Report · Acceptance letter]

7 Sep 2020

PONE-D-19-32152R1 

Challenges to pre-migration interventions to prevent human trafficking: Results from a before-and-after learning assessment of training for prospective female migrants in Odisha, India 

Dear Dr. Pocock:

I'm pleased to inform you that your manuscript has been deemed suitable for publication in PLOS ONE. Congratulations! Your manuscript is now with our production department. 

Kind regards, 

on behalf of

Dr. Michelle L. Munro-Kramer 

Academic Editor

PLOS ONE